# ROBUST CLASSIFICATION VIA REGRESSION FOR LEARNING WITH NOISY LABELS

**Erik Englesson, Hossein Azizpour**
KTH Royal Institute of Technology
`{engless, azizpour}@kth.se`

## ABSTRACT

Deep neural networks and large-scale datasets have revolutionized the field of machine learning. However, these large networks are susceptible to overfitting to label noise, resulting in reduced generalization. To address this challenge, two promising approaches have emerged: i) loss reweighting, which reduces the influence of noisy examples on the training loss, and ii) label correction that replaces noisy labels with estimated true labels. These directions have been pursued separately or combined as independent methods, lacking a unified approach. In this work, we present a unified method that seamlessly combines loss reweighting and label correction to enhance robustness against label noise in classification tasks. Specifically, by leveraging ideas from compositional data analysis in statistics, we frame the problem as a regression task, where loss reweighting and label correction can naturally be achieved with a shifted Gaussian label noise model. Our unified approach achieves strong performance compared to recent baselines on several noisy labelled datasets. We believe this work is a promising step towards robust deep learning in the presence of label noise. Our code is available at: github.com/ErikEnglesson/SGN.

## 1 INTRODUCTION

Deep neural networks (DNNs) fuelled by large-scale labelled datasets such as ImageNet, have achieved remarkable performance in recent years. However, continuing progress with even larger labelled datasets faces a persistent challenge: noisy labels. Even the manually annotated ImageNet dataset contains label noise (Beyer et al., 2020), and using scalable but automatic annotation techniques will lead to higher levels of noise. This poses a problem as DNNs overfit to label noise, which results in suboptimal generalization (Zhang et al., 2017a). Common regularization techniques, including weight decay, help but do not solve the problem. Therefore, designing learning algorithms that are robust to label noise is an important research endeavour.

Current research study several categories of methods for reducing negative impacts of label noise (Han et al., 2020; Algan & Ulusoy, 2021; Song et al., 2022), this includes robust loss functions, sample selection, regularization, among others. In this work, we consider the two prominent directions of loss reweighting and label correction.

*Loss reweighting.* The key idea here is to reduce the contribution of noisy labelled examples to the total loss. This has been shown to effectively delay overfitting to noisy labels. However, intuitively, during later stages of training when most of the correctly labelled examples have been fit, the noisy labels will gradually dominate the gradients, resulting in overfitting, but at a much slower rate.

*Label correction.* The key idea here is to correct the noisy labels by replacing the given labels with estimates of what the ground-truth label is. Clearly, with perfect correction, the problem of noisy labels is solved. Unfortunately, this is easier said than done. A common method is to estimate the true labels during training with DNNs, therefore, intuitively, too early in the learning phase the predictions of the DNN are not accurate, while too late in the training the network has overfit to the noisy labels. Therefore, there is only a short span during learning where the predicted labels are reliable.

We believe these directions are complementary and combining them should be especially fruitful. The proposed high-level approach is simple: we adopt loss reweighting to delay overfitting to noisy labels, so that our label correction procedure has ample time to estimate and correct labels.

These directions have been pursued separately or combined as independent methods, lacking a unified approach. In this work, we present a unified method that seamlessly combines loss reweighting and label correction to enhance robustness against label noise in classification tasks. More precisely, our main contributions are:

- We propose an adaptation of the log-ratio transform approach from compositional data analysis in statistics to the classification task (Section 3.1). This includes turning the classification dataset to a regression dataset (Section 3.2) and to transform regression predictions to classification predictions (Section 3.4).

- With this novel view, we solve the label noise problem in classification as a regression problem. We present a unified probabilistic regression method that seamlessly combines loss reweighting and label correction (Section 3.3). We naturally achieve loss reweighting by learning the mean and covariance of per-example Gaussian distributions. To achieve label correction, we naturally extend this approach by using a shifted (non-zero mean) Gaussian noise model, which we show indirectly changes the label.

- Finally, we perform extensive experiments, that increases our understanding of the method and shows its strong performance compared to baselines on several datasets (Section 4).

## 2 COMPOSITIONAL DATA ANALYSIS WITH LOG-RATIO TRANSFORMS

In this section, we provide background information about the statistical field of compositional data analysis (Aitchison, 2011), and in particular the log-ratio transform approach.

**Compositional Data**   Compositional data is a collection of compositions, which are non-negative real vectors that sum to a constant, usually 1. Indeed, in this case, compositions are categorical distributions in the probability simplex $\Delta^{D-1} = \{[p_1, p_2, \dots, p_D] \in \mathbb{R}^D \mid p_i \geq 0, \sum_{i=1}^{D} p_i = 1\}$ where $D$ denotes the number of components. Compositional data naturally arises in many fields of study (Aitchison, 2005; Tsagris & Stewart, 2020), e.g., in geology when studying [sand, silt, clay] compositions of sedimentary rock. As compositions are constrained variables, compositional data cannot be analysed with statistical techniques designed for unconstrained variables (Pawlowsky-Glahn & Egozcue, 2006). Next, we describe a suitable method for analysing compositional data.

**The Log-Ratio Transform Approach**   Aitchison (1982) recognized that all relevant information for compositions is captured in the relations (ratios) between components, and proposed a framework for analysing compositional data in terms of ratios. The core idea is: i) use a log-ratio transform to view compositions in the constrained simplex space as vectors in an unconstrained multivariate real space, ii) apply standard statistical techniques for unconstrained variables, iii) use the inverse transform to map results back to the simplex. Statistical modelling of compositional data typically assume that the transformed data are normally distributed, leading to a Logistic-Normal distribution (Atchison & Shen, 1980). Clearly, in this approach, the transform has a key role, several of which we will describe next.

**Transforms**   There are many transforms that can be used to transform compositional data to a real space, see *e.g.* (Alenazi, 2023). In this work, we focus on the most common log-ratio transforms. Here, we provide intuition for these transforms, and leave details to the work of Egozcue et al. (2011).

The *additive log-ratio (alr) transform* (Aitchison, 1982) calculates the logarithm of the ratio of $D-1$ components with the remaining component: $alr(p) = \log([p_1, p_2, \dots, p_{D-1}]/p_D)$. The choice of which component to divide by is arbitrary and introduces an asymmetry.

The *centered log-ratio (clr) transform* (Aitchison, 1983) solves the issue of asymmetry by dividing the components of $p$ by its geometric mean $g(p)$: $clr(p) = \log([p_1, p_2, \dots, p_D]/g(p))$. The transformed clr data sum to zero and are therefore constrained to a $D-1$ dimensional hyper-plane of $\mathbb{R}^D$.

The *isometric log-ratio (ilr) transform* (Egozcue et al., 2003) reparametrizes the clr transformed data in an orthonormal bases of the $D-1$ dimensional hyper-plane: $ilr(p) = V^T clr(p)$, where $V$ is a $D \times D-1$ dimensional matrix providing the change of basis. Although the ilr transform is symmetric and its image is the full $\mathbb{R}^{D-1}$, it shares the problem of zero components with other log-

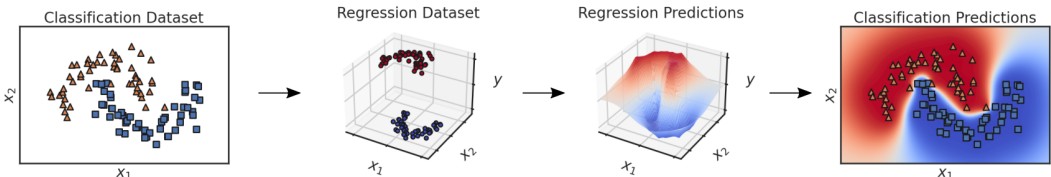

Figure 1: **Method Overview.** Our method is a three-step process: i) classification labels are transformed to regression labels, ii) the regression task is robustly solved with our loss reweighting and label correction method, iii) regression predictions are transformed to classification predictions.

ratio transforms, as the logarithm of zero is undefined. In this work, we use the simple label smoothing technique (Szegedy et al., 2016) to solve this issue, but several other solutions exist (Alenazi, 2023).

## 3    METHOD

In this section, we first provide an overview of our idea to adapt the three-step log-ratio transform approach for classification (Section 3.1). The following sections go over these three steps in detail: i) transforming the classification dataset to a regression dataset (Section 3.2), ii) our regression learning algorithm with loss reweighting and label correction (Section 3.3), iii) transforming regression predictions to classification predictions (Section 3.4).

### 3.1    OVERVIEW: THE LOG-RATIO TRANSFORM APPROACH FOR CLASSIFICATION TASKS

To achieve a unified approach with loss reweighting and label correction for classification tasks, we propose the following three-step process based on the log-ratio transform approach (Section 2):

**Step 1.**    We turn the classification dataset to a compositional dataset via label smoothing. Then, the compositional dataset is transformed to a regression dataset via the ilr transform.

$$\mathcal{D}_\eta^{class} = \{(\boldsymbol{x}_i, y_i)\}_{i=1}^N \to \mathcal{D}_\eta^{comp} = \{(\boldsymbol{x}_i, LS(y_i))\}_{i=1}^N \to \mathcal{D}_\eta^{reg} = \{(\boldsymbol{x}_i, ilr(LS(y_i)))\}_{i=1}^N \quad (1)$$

**Step 2.** Given the regression dataset $\mathcal{D}_\eta^{reg}$, we use our robust regression algorithm that achieves loss reweighting and label correction via a shifted ($\boldsymbol{\Delta}$) Gaussian noise model.

$$ilr(LS(y_i)) = \boldsymbol{t}(\boldsymbol{x}_i) = \boldsymbol{\mu}(\boldsymbol{x}_i) + \boldsymbol{\epsilon}(\boldsymbol{x}_i), \quad \boldsymbol{\epsilon}(\boldsymbol{x}_i) \sim \mathcal{N}(\boldsymbol{\Delta}(\boldsymbol{x}_i), \boldsymbol{\Sigma}(\boldsymbol{x}_i)) \quad (2)$$

**Step 3.** We transform any regression prediction $\hat{\boldsymbol{\mu}} \in \mathbb{R}^{K-1}$, to the probability simplex with the inverse ilr transform, and use the most likely class as the class prediction

$$\hat{\boldsymbol{\mu}} \quad \to \quad \hat{\boldsymbol{\pi}} = ilr^{-1}(\hat{\boldsymbol{\mu}}) \quad \to \quad \hat{y} = \arg\max_k \hat{\boldsymbol{\pi}}_k \quad (3)$$

An overview of our approach is shown in Figure 1. Next, we go over each of these steps in detail.

### 3.2    STEP 1: TRANSFORMING CLASSIFICATION DATASETS TO REGRESSION DATASETS

The log-ratio transform approach transforms compositional datasets to regression datasets. Hence, we can get regression datasets by converting classification datasets to compositional datasets. Next, we discuss using label smoothing for this purpose and provide details about the ilr transform we use.

**Class ID to Interior Simplex**    It is common to view the class ID labels in classification as part of the simplex via the one-hot encoding, effectively turning the classification dataset into a compositional dataset. Unfortunately, this is not enough for our purposes, as the log-ratio transforms cannot deal with zero components. To ensure that all components are non-zero (interior of simplex), we simply use the well-known label smoothing (Szegedy et al., 2016) technique, which is defined as

$$LS(y) = (1 - \gamma)\boldsymbol{\delta}_y + \gamma\boldsymbol{u} \quad (4)$$

where $\boldsymbol{\delta}_y \in \Delta^{K-1}$ is a one-hot encoding of $y$ (component y is 1 while the rest are zero), and $\boldsymbol{u}$ is the uniform distribution over $K$ classes (all components equal to $1/K$), and $\gamma$ is a small positive number. We motivate the use of label smoothing as it is simple to understand, implement, and has been shown to help against label noise for the standard cross-entropy loss (Lukasik et al., 2020; Wei et al., 2022).

**The Isometric Log-Ratio Transform**  In Section 2, we described the additive, centered, and the isometric log-ratio transforms. In this work, we use the ilr transform (Egozcue et al., 2003), as it does not have some issues that the other transforms have: asymmetry (alr) and the image being on a lower dimensional plane (clr). To use the ilr transform, one has to choose an orthonormal basis and its corresponding matrix $V$. Here, as originally proposed by Egozcue et al. (2003), we have $V^t$ be a $K \times K$ dimensional Helmert matrix in the strict sense (Lancaster, 1965) without its first row. The columns of this $K \times K - 1$ matrix $V$ correspond to the orthogonal basis vectors of the clr plane.

### 3.3 STEP 2: REGRESSION ANALYSIS WITH LOSS REWEIGHTING AND LABEL CORRECTION

With label smoothing and the ilr transform, we can transform the classification task into a regression task, allowing us to use classical regression techniques. Here, we first describe a statistical regression technique that naturally achieves loss attenuation and then uncover its potential to do label correction as well. This is then followed by details on how to achieve this label correction in practice.

**Loss Reweighting via a Gaussian Noise Model**  Nix & Weigend (1994) proposed to model the observed regression target $t$ as having a true mean $\mu$ with some added noise $\epsilon$: $t(x) = \mu(x) + \epsilon(x)$, where $\epsilon$ is normally distributed with zero mean and input-dependent variance $\sigma^2(x)$. This results in a Gaussian likelihood $p(t|x) = \mathcal{N}(\mu(x), \sigma^2(x))$ per example. The authors propose to have a neural network with two output heads to output $\hat{\mu} = \mu_{\boldsymbol{\theta}}$ and $\hat{\sigma}^2 = \sigma_{\boldsymbol{\theta}}^2$ per example, and the parameters of the network $\boldsymbol{\theta}$ are found through maximum likelihood estimation. That is

$$\arg\min_{\boldsymbol{\theta}} -\sum_{i=1}^{N} \log \mathcal{N}(t_i; \mu_{\boldsymbol{\theta}}(x_i), \sigma_{\boldsymbol{\theta}}^2(x_i)) = \arg\min_{\boldsymbol{\theta}} \sum_{i=1}^{N} \frac{(t_i - \mu_{\boldsymbol{\theta}}(x_i))^2}{2\sigma_{\boldsymbol{\theta}}^2(x_i)} + \frac{1}{2}\log \sigma_{\boldsymbol{\theta}}^2(x_i) \quad (5)$$

Kendall & Gal (2017) recognized that this objective, when learning $\sigma_{\boldsymbol{\theta}}^2(x_i)$ does loss reweighting (they called it attenuation). Clearly, if the variance $\sigma_{\boldsymbol{\theta}}^2(x_i)$ is high for noisy examples, their influence on the total loss is reduced. Here, the last term of the objective in Equation 5 naturally arises from the probabilistic noise model and importantly keeps the network from outputting the trivial solution of high variance (low weight) for all examples. In this work, we use this approach but in a multivariate setting using deep neural networks to estimate the mean $\boldsymbol{\mu}_{\boldsymbol{\theta}}$ and covariance $\Sigma_{\boldsymbol{\theta}}$ per-example. To efficiently and effectively parameterize covariance matrices, we follow the works of Collier et al. (2021); Englesson et al. (2023), see Appendix A.1 for details.

**Uncovering Label Correction via a Shifted Gaussian Noise Model**  The Gaussian noise model above achieves loss reweighting, but not yet label correction. Unfortunately, loss reweighting is not enough for robustness as the ML estimate for $\mu_{\boldsymbol{\theta}}$ is $\hat{\mu}_{ML}(x_i) = t_i$ for all $i = 1, \ldots, N$, *i.e.* it would fit all (including noisy) labels. Here, we propose the following shifted noise model: $t = \mu + \epsilon, \epsilon \sim \mathcal{N}(\Delta, \sigma^2)$, that indirectly changes the label in the training objective

$$
\begin{aligned}
-\log \mathcal{N}(t; \mu_{\boldsymbol{\theta}} + \Delta, \sigma_{\boldsymbol{\theta}}^2) &= \frac{(t - (\mu_{\boldsymbol{\theta}} + \Delta))^2}{2\sigma_{\boldsymbol{\theta}}^2} + \frac{1}{2}\log \sigma_{\boldsymbol{\theta}}^2 \\
&= \frac{((t - \Delta) - \mu_{\boldsymbol{\theta}})^2}{2\sigma_{\boldsymbol{\theta}}^2} + \frac{1}{2}\log \sigma_{\boldsymbol{\theta}}^2 \\
&= -\log \mathcal{N}(t - \Delta; \mu_{\boldsymbol{\theta}}, \sigma_{\boldsymbol{\theta}}^2)
\end{aligned}
\tag{6}
$$

Consider an example with a noisy target $t$ where the correct label is $\mu$. We can correct the label by having $t - \Delta = \mu$, or, equivalently, $\Delta = t - \mu$. Therefore, including a shift $\Delta$ in the noise model indirectly incorporates label correction. Although intuitively simple, how to estimate the shift is far from trivial, for which a practical algorithm is described next.

**A Practical Algorithm to Estimate the Shift**  To do label correction with $\Delta$, we want $\Delta = t - \mu$. As the observed target $t$ is known, it is enough to estimate the ground-truth label $\mu$ and then calculate the difference. Clearly, this is a challenging task, since if we were able to accurately estimate $\mu$, the problem of noisy labels would be solved. To achieve good estimates of ground-truth labels, we rely on the predictions of a network with moving averaged weights of the original network from previous training steps. Despite being a simple approach, these networks have been shown to generalize better (Izmailov et al., 2018), and have been successfully used in semi-supervised (Tarvainen & Valpola, 2017) and self-supervised (Grill et al., 2020) learning methods.

In particular, for each training example $(x_i, t_i)$, we estimate $\Delta_{\bar{\boldsymbol{\theta}}}(x_i) = t_i - \mu_{\bar{\boldsymbol{\theta}}}(x_i)$, where $\mu_{\bar{\boldsymbol{\theta}}}$ is the prediction of the network with exponentially moving averaged (EMA) weights $\bar{\boldsymbol{\theta}}$ of the original network parameters $\boldsymbol{\theta}$. Substituting $\Delta$ with $\Delta_{\bar{\boldsymbol{\theta}}}$ in Equation 6, we get

$$-\log\mathcal{N}(t; \mu_{\boldsymbol{\theta}} + \Delta_{\bar{\boldsymbol{\theta}}}, \sigma_{\boldsymbol{\theta}}^2) = -\log\mathcal{N}(\mu_{\bar{\boldsymbol{\theta}}}; \mu_{\boldsymbol{\theta}}, \sigma_{\boldsymbol{\theta}}^2) \tag{7}$$

That is, the introduction of the shift $\Delta_{\bar{\boldsymbol{\theta}}}$ effectively changes the label from the observed $t$ to the estimated ground-truth $\mu_{\bar{\boldsymbol{\theta}}}$. This is a desired capability of any label correction method, but it also introduces a problem early in training, as the EMA network's predictions are poor estimates of the ground-truth label at this point. To remedy this, we could start introducing $\Delta_{\bar{\boldsymbol{\theta}}}$ after a few warm-up epochs, as proposed by Grill et al. (2020). However, instead of abruptly changing the label in this way, we use a smooth transition using the following weighting scheme: $t = \mu + (1 - \alpha^e)\Delta$, where $\alpha$ is a hyperparameter close to 1 and $e$ is the current training epoch. In this way, the weight $(1 - \alpha^e)$ is close to zero early in training and gradually goes to 1, causing the labels to transition from the observed to the estimated ground-truth labels during the training process.

### 3.4 STEP 3: TRANSFORMING REGRESSION PREDICTIONS TO CLASS IDS

In this section, we complete the method by considering the last step: converting these predictions back to classification predictions.

**Noise-Free Predictions**  The networks output the estimated true mean $\hat{\mu}$ and the estimated noise variance $\hat{\sigma}^2$. For predictions on the training set, this noise variance is important to achieve loss reweighting, but for unseen data our goal is to predict the noise-free true mean. Therefore, at test time, we do not use the predicted $\hat{\sigma}^2$ and consider the estimated true mean $\hat{\mu}$ as the prediction of the network. As the ilr transform is one-to-one, we utilize its inverse to map $\hat{\mu}$ to a categorical distribution in the simplex: $\hat{\boldsymbol{\pi}} = ilr^{-1}(\hat{\mu})$. To obtain a class ID prediction, the standard approach of using the most likely component is used: $\hat{y} = \arg\max_k \hat{\boldsymbol{\pi}}_k$.

**EMA Predictions**  A strong advantage of estimating the ground-truth target with an EMA network compared to, *e.g.*, with the same network (Reed et al., 2014) or with a separate buffer (Liu et al., 2020), is that we have the option to use the predictions of this network at no extra cost at test time. As the EMA network has been shown to generalize better (Izmailov et al., 2018), this is a natural choice. We perform an ablation study to quantify the improvements this leads to in Section 4.4.

## 4 EXPERIMENTS

First, we describe our experimental setup (Section 4.1), followed by results on synthetically (Section 4.2) and naturally (Section 4.3) noisy datasets. Finally, we perform experiments to better understand our method (Section 4.4). For details about the CIFAR, CIFAR-N, Clothing1M, and WebVision datasets, see Appendix A.4. We refer to our method as Shifted Gaussian Noise (SGN).

### 4.1 EXPERIMENTAL SETUP

For the experiments on the CIFAR (including CIFAR-N) datasets, we implement all baselines in the same shared code base to have an as conclusive comparison as possible. To achieve the best possible performance in this setup, we do a search for method-specific hyperparameters for each method based on a noisy validation set. For more information about the search and the optimal hyperparameters for each method, see Appendix A.3. For full details of the training setup on the CIFAR, CIFAR-N, Clothing1M, and WebVision datasets, see Appendix A.2.

The results on all datasets report the mean and standard deviation of validation/test accuracy calculated based on five different network training runs with different seeds, evaluated at the end of training, if not stated otherwise. The random seeds affect both network initialization, synthetic noise generation, data augmentation, and the order of the data loaders.

### 4.2 SYNTHETIC LABEL NOISE EXPERIMENTS

To evaluate the robustness of our approach in a setup where we have full control over the amount of label noise, we perform experiments on the CIFAR datasets with synthetic noise manually added to

Table 1: **Synthetic Noise: CIFAR-10 and CIFAR-100.** All methods are implemented in a common code base, and hyperparameters are searched for. We report the mean and standard deviation of five different runs, where results in bold has no statistically significant difference compared to the method with the highest mean accuracy. Our method consistently demonstrates strong robustness compared to the baselines across various noise rates, noise types and datasets.

| | Method | No Noise | Symmetric Noise Rate | | | Asymmetric Noise Rate | | |
|---|---|---|---|---|---|---|---|---|
| | | 0% | 20% | 40% | 60% | 20% | 30% | 40% |
| CIFAR-10 | CE | 90.67 ± 0.80 | 73.54 ± 1.01 | 56.56 ± 1.44 | 39.44 ± 1.87 | 81.35 ± 1.26 | 76.01 ± 2.67 | 71.89 ± 1.67 |
| | GCE | 90.83 ± 0.44 | 87.55 ± 0.41 | 84.72 ± 0.82 | 79.25 ± 0.93 | 85.68 ± 0.69 | 83.97 ± 0.52 | 72.90 ± 1.61 |
| | LS | 89.78 ± 0.39 | 79.09 ± 0.96 | 64.27 ± 1.50 | 43.57 ± 3.13 | 81.99 ± 1.22 | 76.49 ± 1.17 | 71.66 ± 1.78 |
| | HET | 90.82 ± 0.42 | 77.16 ± 0.94 | 62.85 ± 1.88 | 44.20 ± 3.05 | 81.55 ± 0.80 | 77.05 ± 0.33 | 72.69 ± 0.89 |
| | NAN | 89.61 ± 0.93 | 83.86 ± 1.03 | 79.80 ± 0.59 | 73.58 ± 0.41 | 84.32 ± 1.05 | 76.79 ± 2.28 | 72.90 ± 1.92 |
| | LN | 90.17 ± 0.55 | 86.13 ± 1.03 | 81.37 ± 1.97 | 76.08 ± 0.63 | 87.64 ± 0.78 | 86.91 ± 1.03 | 82.18 ± 1.30 |
| | ELR | 91.78 ± 0.26 | 90.15 ± 0.54 | 88.19 ± 0.68 | 81.87 ± 2.42 | 90.59 ± 0.36 | 89.72 ± 0.22 | 87.37 ± 0.55 |
| | SOP | 91.57 ± 0.38 | 89.86 ± 0.45 | 88.45 ± 0.51 | **85.56 ± 0.93** | 89.84 ± 0.55 | 87.60 ± 0.65 | 83.90 ± 1.04 |
| | NAL | 92.80 ± 0.23 | 89.79 ± 0.47 | 86.25 ± 0.28 | 78.82 ± 0.43 | 90.80 ± 0.76 | 89.51 ± 0.76 | 84.36 ± 1.19 |
| | SGN (Ours) | **94.12 ± 0.22** | **93.02 ± 0.17** | **91.29 ± 0.25** | **86.03 ± 1.19** | **93.35 ± 0.21** | **92.71 ± 0.11** | **91.26 ± 0.27** |
| CIFAR-100 | CE | 64.87 ± 0.88 | 47.39 ± 0.43 | 33.62 ± 0.79 | 20.04 ± 0.58 | 50.98 ± 0.88 | 44.04 ± 0.73 | 36.95 ± 0.58 |
| | GCE | 64.33 ± 0.83 | 61.67 ± 0.67 | 53.96 ± 1.40 | 42.85 ± 0.79 | 59.63 ± 1.28 | 49.21 ± 0.53 | 36.78 ± 0.50 |
| | LS | 65.39 ± 0.40 | 57.08 ± 0.70 | 44.03 ± 1.20 | 26.13 ± 1.45 | 55.47 ± 0.76 | 44.70 ± 0.73 | 38.56 ± 0.66 |
| | HET | 65.18 ± 0.90 | 54.83 ± 0.46 | 41.49 ± 1.53 | 22.42 ± 0.95 | 61.29 ± 0.46 | 56.44 ± 0.53 | 45.75 ± 1.02 |
| | NAN | 64.25 ± 0.64 | 56.93 ± 0.77 | 50.03 ± 0.62 | 40.45 ± 0.41 | 56.40 ± 1.07 | 52.78 ± 0.85 | 40.59 ± 0.84 |
| | LN | 64.88 ± 0.98 | 60.58 ± 1.07 | 55.55 ± 1.30 | 46.43 ± 1.15 | 64.31 ± 0.98 | 64.07 ± 0.77 | 61.20 ± 1.22 |
| | ELR | 67.74 ± 0.61 | 64.70 ± 0.85 | 59.92 ± 0.95 | 48.85 ± 0.85 | 66.32 ± 0.88 | 65.99 ± 1.16 | 63.80 ± 0.35 |
| | SOP | 62.50 ± 0.76 | 61.40 ± 1.18 | 60.92 ± 1.34 | 50.80 ± 0.74 | 54.19 ± 0.48 | 47.22 ± 1.27 | 39.20 ± 0.60 |
| | NAL | 69.59 ± 0.37 | 64.27 ± 0.18 | 57.09 ± 0.51 | 46.23 ± 0.45 | 66.59 ± 0.48 | 64.46 ± 0.62 | 58.01 ± 0.79 |
| | SGN (Ours) | **73.88 ± 0.34** | **71.79 ± 0.26** | **66.86 ± 0.35** | **56.83 ± 0.57** | **72.83 ± 0.31** | **72.16 ± 0.86** | **71.01 ± 0.71** |

a given percentage of training examples. Here, we describe our process of adding label noise, the baselines we compare with, and finally our experimental results.

**Label Noise Process** We consider two types of synthetic noise: symmetric and asymmetric. Symmetric noise uniformly samples a new label from any class. Asymmetric noise circularly cycle labels to the next class on CIFAR-100, and as follows on CIFAR-10: bird → airplane, cat ↔ dog, deer → horse, and truck → automobile. For more details on these noise types, see Appendix A.11.

**Baselines** We compare our novel method with an extensive set of baselines, from the standard cross-entropy loss, to strong representatives from the categories of regularization, robust loss functions, loss reweighting and label correction methods. We compare with the robust GCE (Zhang & Sabuncu, 2018) loss and ELR (Liu et al., 2020) that regularizes the prediction to be closer to the estimated

Table 2: **Natural Noise: CIFAR-N.** All methods are implemented in a common code base, and hyperparameters are searched for. We report the mean and standard deviation of five different runs, where results in bold has no statistically significant difference compared to the method with the highest mean accuracy. Our method consistently demonstrates strong robustness across all settings.

| Method | CIFAR-10N | | | | | CIFAR-100N |
|---|---|---|---|---|---|---|
| | Aggregate | Random 1 | Random 2 | Random 3 | Worst | |
| CE | 83.59 ± 0.98 | 77.75 ± 0.74 | 75.52 ± 1.08 | 76.25 ± 1.26 | 59.01 ± 0.98 | 42.75 ± 0.93 |
| GCE | 86.66 ± 0.68 | 85.66 ± 0.73 | 85.58 ± 0.65 | 84.78 ± 0.62 | 77.48 ± 1.22 | 48.81 ± 0.46 |
| LS | 85.08 ± 0.54 | 80.07 ± 0.82 | 79.81 ± 0.62 | 79.41 ± 0.68 | 63.07 ± 1.93 | 45.98 ± 1.44 |
| HET | 84.45 ± 0.57 | 78.87 ± 0.47 | 76.24 ± 0.96 | 77.68 ± 1.93 | 63.27 ± 2.62 | 45.58 ± 0.80 |
| NAN | 85.53 ± 0.83 | 81.85 ± 1.13 | 83.40 ± 0.84 | 82.77 ± 0.78 | 75.47 ± 0.76 | 50.00 ± 0.72 |
| LN | 85.35 ± 1.33 | 83.70 ± 0.80 | 83.65 ± 0.82 | 84.07 ± 0.71 | 74.31 ± 1.08 | 50.37 ± 0.50 |
| ELR | 89.61 ± 0.12 | 89.05 ± 0.89 | 88.79 ± 0.72 | 88.88 ± 0.61 | 82.59 ± 0.54 | 54.91 ± 1.11 |
| SOP | 89.54 ± 0.57 | 89.65 ± 0.79 | 89.43 ± 0.56 | 89.54 ± 0.39 | 82.17 ± 0.90 | 50.20 ± 0.84 |
| NAL | 91.30 ± 0.16 | 89.22 ± 0.60 | 89.09 ± 0.24 | 89.22 ± 0.39 | 81.39 ± 1.22 | 56.33 ± 1.21 |
| SGN (Ours) | **92.06 ± 0.12** | **91.94 ± 0.19** | **91.69 ± 0.22** | **91.91 ± 0.10** | **86.67 ± 0.42** | **60.36 ± 0.71** |

ground-truth label. As label smoothing (LS) increases robustness to label noise (Lukasik et al., 2020), and our method uses it to get to a compositional dataset, it is a natural baseline. Furthermore, as our method uses a Gaussian label noise model, we compare with NAN (Chen et al., 2020) and HET (Collier et al., 2021) that adds Gaussian noise to the one-hot encoded labels and pre-softmax logits, respectively. Finally, we also compare with methods that have some form of loss reweighting and or label correction, including SOP (Liu et al., 2022), NAL (Lu et al., 2022) and LN (Englesson et al., 2023). Detailed connections to the most related works are discussed in Appendix A.6.

**Results** In Table 1, we show the results for various levels of symmetric and asymmetric noise on CIFAR-10 and CIFAR-100. Our method (SGN) performs significantly better than other methods on both datasets for all noise rates (except 60% symmetric noise) and noise types. Interestingly, SGN significantly outperforms CE when there is no added label noise. We attribute this to better generalization of the EMA network, and to inherent label noise in the datasets, as other label estimating method often also perform better. Remarkably, comparing SGN on no noise with 40% asymmetric noise, the generalization degradation is less than 3% on both datasets, which we attribute to an effective label correction, see Section 4.4. The most challenging setup for all methods seem to be high levels of symmetric noise. In our case, we believe this is mainly due to our method predicting the covariance matrices of the Gaussian distributions. As there is no structure in the label noise, the network has to memorize the noise to do an effective loss reweighting (Englesson et al., 2023). Despite this, SGN with 40% noise remarkably outperforms CE when trained with 0% added noise.

## 4.3 NATURAL LABEL NOISE EXPERIMENTS

Synthetic label noise is excellent for conducting experiments with controlled levels of label noise. However, this comes at the cost of the structure of the noise potentially being different compared to label noise naturally arising from standard annotation processes. In this section, we turn our attention to this type of noise by conducting experiments on datasets that have been annotated by humans, or automatically with search engines.

**Results** We report the results on CIFAR-10N (Aggregate, Random 1-3, and Worst label sets) and CIFAR-100N datasets in Table 2. Our method shows strong robustness in all six cases. We find that our method achieves the largest improvements compared to baselines on the most challenging setups, *i.e.*, "Worst" (40% noise) on CIFAR-10N and CIFAR-100N. For example, on CIFAR-100N, our method achieves a four percentage point improvement compared to the best baseline (NAL). However, ELR, SOP and NAL baselines are all showing strong performance on these datasets.

In Table 3 we report strong results on Clothing1M and WebVision. All baseline results (except NAL) are gathered from the work of Liu et al. (2022). These results correspond to accuracies of a single network evaluated when the best validation accuracy is achieved. NAL results are from Lu et al. (2022), but has a different evaluation setup on WebVision.[1] We provide results following our standard evaluation setup, *i.e.*., mean and standard deviation at the end of training. We believe the use of early

---

[1]The official code evaluates all accuracies in each epoch and tracks the best accuracies separately in training.

Table 3: **Natural Noise: Clothing1M & (mini) WebVision.** All baseline results (except NAL) are from the work of Liu et al. (2022). We follow their evaluation setup of doing early stopping, and also report mean and standard deviation at the end of training (†). NAL results are from Lu et al. (2022), but they follow a different evaluation setup on WebVision, thus marked with *.

| Method | Clothing1M | WebVision Validation | | WebVision Test (ILSVRC12) | |
|---|---|---|---|---|---|
| | Top 1 | Top 1 | Top 5 | Top 1 | Top 5 |
| Forward | 69.8 | 61.1 | - | 57.3 | - |
| Co-teaching | 69.2 | 63.6 | - | 61.5 | - |
| ELR | 72.9 | 76.3 | 91.3 | 68.7 | 87.8 |
| SOP | 73.5 | 76.6 | - | 69.1 | - |
| NAL | 73.6 | 77.4* | 92.3* | 74.1* | 92.1* |
| SGN (Ours) | 73.9 | 77.2 | 91.2 | 72.6 | 90.5 |
| SGN† (Ours) | 73.6 ± 0.27 | 76.12 ± 0.36 | 90.74 ± 0.29 | 72.72 ± 0.17 | 90.35 ± 0.28 |

Table 4: **Ablation Study.** We report results on synthetic and natural noise on the CIFAR datasets when we systematically deactivate the three components of the method: loss reweighting (LR), label correction (LC), and EMA prediction (EP). Synthetic noise rates are all at 40%.

| LR | LC | EP | CIFAR-10 | | CIFAR-10N | | CIFAR-100 | | CIFAR-100N |
|----|----|----|----------|----------|-----------|----------|-----------|----------|-----------|
| | | | Symmetric | Asymmetric | Random 1 | Worst | Symmetric | Asymmetric | |
| ✗ | ✗ | ✗ | $65.51 \pm 2.43$ | $71.73 \pm 1.35$ | $80.47 \pm 1.52$ | $63.05 \pm 1.38$ | $47.87 \pm 1.28$ | $37.75 \pm 0.79$ | $47.29 \pm 0.67$ |
| ✓ | ✗ | ✗ | $77.62 \pm 0.59$ | $78.15 \pm 0.58$ | $83.47 \pm 0.81$ | $72.58 \pm 1.08$ | $53.54 \pm 0.83$ | $59.03 \pm 1.12$ | $48.61 \pm 1.24$ |
| ✓ | ✓ | ✗ | $87.66 \pm 0.68$ | $86.49 \pm 0.83$ | $88.67 \pm 1.09$ | $84.24 \pm 0.28$ | $61.10 \pm 0.75$ | $63.80 \pm 1.16$ | $55.72 \pm 0.65$ |
| ✗ | ✗ | ✓ | $73.12 \pm 0.64$ | $76.15 \pm 0.51$ | $87.04 \pm 0.32$ | $71.64 \pm 0.42$ | $57.06 \pm 0.44$ | $44.22 \pm 0.35$ | $55.18 \pm 0.28$ |
| ✓ | ✗ | ✓ | $84.86 \pm 0.60$ | $83.97 \pm 0.25$ | $89.04 \pm 0.13$ | $80.30 \pm 0.33$ | $64.04 \pm 0.47$ | $67.76 \pm 1.30$ | $57.25 \pm 0.48$ |
| ✓ | ✓ | ✓ | $\mathbf{91.29 \pm 0.25}$ | $\mathbf{91.26 \pm 0.27}$ | $\mathbf{91.94 \pm 0.19}$ | $\mathbf{86.67 \pm 0.42}$ | $\mathbf{66.86 \pm 0.35}$ | $\mathbf{71.01 \pm 0.71}$ | $\mathbf{60.36 \pm 0.71}$ |

stopping, and no error bars, makes the results noisy and comparisons inconclusive. However, as early stopping improves performance, we also report results following the first setup of Liu et al. (2022). We believe this highlights the importance of having common training and evaluation setups.

## 4.4 EMPIRICAL ANALYSIS AND DISCUSSIONS

Here, we dissect the impact of different parts of the method, and its label correction behaviour.

**Ablation Study** To better understand our method, we perform an ablation study where we systematically deactivate key components of our method: loss reweighting (LR), label correction (LC), and the use of EMA network predictions (EP). As the covariance in the Gaussian likelihood causes the loss reweighting effect, we can deactivate it by using an identity matrix. As the shift $\Delta$ is responsible for the label correction, we can deactivate it by setting $\alpha = 1$. We can deactivate the EMA predictions by instead using the predictions of the main network. We report results for different settings on the CIFAR and CIFAR-N datasets in Table 4. Without EMA network predictions, we see that activating LR and LC each provides clear improvements on all datasets and noise types. For example, the activation of LR and then LC improve performance by 20 and 4 pp on CIFAR-100 with asymmetric noise, respectively. Similar improvements are observed when EMA predictions are used. We find that in every case, EMA network predictions provide significant generalization improvements. This study provides strong evidence that the components of our method naturally works well together to provide significant and separate generalization improvements.

**Label Correction Analysis** To better understand the label correction part of our method, we compare the predictions of the EMA network with the clean (original) labels. In Figures 2a and 2b, we show the clean training accuracy of the EMA network predictions during training with synthetic label noise on the CIFAR-100 dataset. Our method struggles to correct labels on high levels of symmetric noise, but does remarkably well on asymmetric noise. We attribute this to the network having to memorize the labels to effectively do loss reweighting on the unstructured symmetric

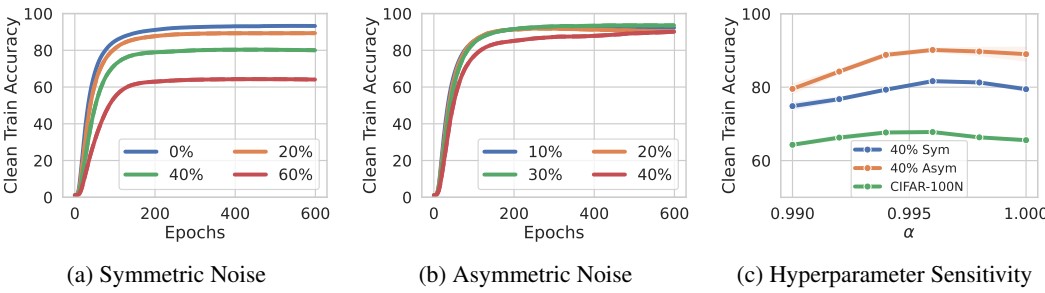

| (a) Symmetric Noise | (b) Asymmetric Noise | (c) Hyperparameter Sensitivity |
|---|---|---|

Figure 2: **Label Correction.** To understand the label correction capabilities of the EMA network, we plot the clean training accuracy during training with various levels of symmetric (a) and asymmetric (b) noise on CIFAR-100. Furthermore, we plot the clean training accuracy at the end of training for various values of the hyperparameter $\alpha$ (c), which affects the scaling of the shift $\Delta$.

noise (Englesson et al., 2023). Despite this, Table 1 shows that our method has the strongest robustness to symmetric noise. We provide similar figures for CIFAR-10 in Section A.5, showing better LC performance. Finally, the 40% curve in Figure 2b shows that it takes time to converge. This, and the flatness of other curves, likely explains why training for longer improves the performance.

To better understand the impact of the gradual introduction of $\Delta$ via the weighting $(1 - \alpha^e)\Delta$ in our noise model, we plot the final clean training accuracy of the EMA network predictions when trained with different values of $\alpha$, see Figure 2c. We find that varying $\alpha$ has a considerable effect on the final accuracy, which is expected as $\alpha = 1$ corresponds to no shift, *i.e.*, turns off LC. Intuitively, a too low $\alpha$ could lead to using the EMA network predictions too early in training when they are not yet accurate. Conversely, a too large $\alpha$ could lead to relying on the noisy labels for too long.

## 5    RELATED WORK

Robustness to label noise is an active area of research, with several important categories of methods. Unfortunately, we cannot cover all of them here, but luckily there are excellent surveys available (Han et al., 2020; Algan & Ulusoy, 2021; Song et al., 2022). Here, we focus on the following categories: regularization, robust loss functions, loss correction, loss reweighting, and label correction.

**Regularization**    A natural idea to try against label noise is standard regularization techniques, which indeed have been shown to help: label smoothing (Szegedy et al., 2016; Lukasik et al., 2020; Wei et al., 2021a), data augmentation (Zhang et al., 2017b), dropout (Rusiecki, 2020; Goel & Chen, 2021), early stopping (Li et al., 2020b; Bai et al., 2021), and adding noise to labels (Chen et al., 2020).

**Robust Loss Functions**    These methods propose replacing the standard CE loss with a robust alternative, often with theoretical guarantees on robustness based on simplifying assumptions, *e.g.*, class-dependent noise. In their seminal work Ghosh et al. (2017) proposed such a theoretical framework that has led to several new loss function (Zhang & Sabuncu, 2018; Wang et al., 2019; Ma et al., 2020; Englesson & Azizpour, 2021), and extension of the theory (Zhou et al., 2021).

**Loss Correction**    Another natural idea is to model the label noise mathematically, by relating the true and noisy label distribution via a transition matrix (Sukhbaatar et al., 2014; Patrini et al., 2017). As for robust loss functions, these methods also have theoretical robustness guarantees.

**Loss Reweighting**    An intuitive idea is to down weight the influence noisy labelled examples has on the training loss, leading to less overfitting. How to estimate such weights has been studied extensively (Wang et al., 2017b;a; Jiang et al., 2018; Ren et al., 2018; Thulasidasan et al., 2019; Lu et al., 2022). To avoid down weighting all examples, an extra loss term is often designed. Interestingly, modelling label noise in the pre-softmax logit space neatly solves this problem (Kendall & Gal, 2017; Collier et al., 2021). In the same direction, Englesson et al. (2023) proposes to maximize a Logistic-Normal likelihood, showing it exhibits loss reweighting properties. In fact, this loss reweighting method corresponds to a special case of our adapted log-ratio transform approach.

**Label Correction**    Here, the idea is to directly fix the cause of the issues: the noisy labels. In fact, a common issue with most of the above methods is that the global optima of the training loss is still to match all (including noisy) labels. Estimating ground-truth labels is an ambitious goal, but often hard to do in practice. Some methods do so with the same network during training (Reed et al., 2014; Song et al., 2019; Zhang et al., 2021), via averages of previous predictions (Laine & Aila, 2017; Liu et al., 2020; Lu et al., 2022), learned via buffers (Liu et al., 2022), or via meta-learning (Shu et al., 2019; Zhang et al., 2020; Shu et al., 2023). Instead of providing a single corrected label distribution, a set of candidates can be used instead as in label relaxation (Lienen & Hüllermeier, 2021).

## 6    CONCLUSION

Our goal was to achieve robustness to label noise in classification via regression-based loss reweighting and label correction. To achieve this, we proposed to adapt the log-ratio transform approach from compositional data analysis in statics to the classification task, allowing us to use regression techniques to solve the problem. Consequently, we successfully achieved our goal by also proposing a regression method based on a shifted Gaussian label noise model that naturally achieved loss reweighting and label correction. We performed extensive experiments, which increased our understanding of our novel method and showed its strong robustness against label noise. We believe this work provides a promising direction towards robust deep learning in the presence of label noise.

**Acknowledgement**. This work was partially supported by the Wallenberg AI, Autonomous Systems and Software Program (WASP) funded by the Knut and Alice Wallenberg Foundation. All experiments were performed using the supercomputing resource Berzelius provided by the National Supercomputer Centre at Linköping University and the Knut and Alice Wallenberg foundation.

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

# A APPENDIX

## A.1 ESTIMATING THE MEAN AND COVARIANCE

A practical way to estimate the covariance matrix of a Gaussian distribution has been proposed by Collier et al. (2021); Englesson et al. (2023). We follow this approach and have a two-headed deep neural network predict $\boldsymbol{\mu_\theta}$ and also an efficient low-rank approximation of a covariance matrix, where the covariance is represented as follows

$$\boldsymbol{\Sigma_\theta}(x) = \boldsymbol{S_\theta}(x)\boldsymbol{S_\theta}(x)^T \tag{8}$$

$$\boldsymbol{S_\theta}(x) = \boldsymbol{r_\theta}(x)\boldsymbol{r_\theta}(x)^T + \boldsymbol{I} \tag{9}$$

where $\boldsymbol{r_\theta} \in \mathbb{R}^{K-1}$ is outputted by the network. A benefit of this particular parametrization is that it can be evaluated efficiently. To see this, we first rewrite the negative log-likelihood of a multivariate distribution in terms of the scale matrix $\boldsymbol{S_\theta}(x)$ instead of $\boldsymbol{\Sigma_\theta}(x)$

$$-\log \mathcal{N}(\boldsymbol{t}(x); \boldsymbol{\mu_\theta}(x), \boldsymbol{\Sigma_\theta}(x)) = \frac{1}{2}(\boldsymbol{t}(x) - \boldsymbol{\mu_\theta}(x))^T \boldsymbol{\Sigma_\theta}^{-1}(x)(\boldsymbol{t}(x) - \boldsymbol{\mu_\theta}(x)) + \frac{1}{2}\log|\boldsymbol{\Sigma_\theta}(x)| + const \tag{10}$$

$$= \frac{1}{2}(\|\boldsymbol{S_\theta}^{-1}(x)(\boldsymbol{t}(x) - \boldsymbol{\mu_\theta}(x))\|_2^2) + \log|\boldsymbol{S_\theta}(x)| + const \tag{11}$$

To evaluate this loss, we need to calculate the inverse and determinant of $\boldsymbol{S}$. In general, this is computationally challenging, but with this particular parameterization we can efficiently calculate the inverse and determinant using the Woodbury matrix identity, and the matrix determinant lemma (see Equations 156 and 24 in Petersen et al. (2008)), respectively.

$$\boldsymbol{S_\theta}^{-1}(x) = (\boldsymbol{r_\theta}(x)\boldsymbol{r_\theta}(x)^T + \boldsymbol{I})^{-1} = I - \frac{\boldsymbol{r_\theta}(x)\boldsymbol{r_\theta}(x)^T}{1 + \boldsymbol{r_\theta}(x)^T \boldsymbol{r_\theta}(x)} \tag{12}$$

$$|\boldsymbol{S_\theta}^{-1}(x)| = |\boldsymbol{r_\theta}(x)\boldsymbol{r_\theta}(x)^T + \boldsymbol{I}| = 1 + \boldsymbol{r_\theta}(x)^T \boldsymbol{r_\theta}(x) \tag{13}$$

Thus, the inverse and determinant computations reduce to simple matrix additions and inner and outer products of vectors, which can be computed efficiently. In practice, we conveniently make use of these loss evaluation optimizations by calling the "log_prob" method of the multivariate normal distribution (MultivariateNormalDiagPlusLowRank) in TensorFlow Probability (Dillon et al., 2017).

## A.2 TRAINING DETAILS

**CIFAR and CIFAR-N** All methods use the same WideResNet (WRN-28-2) architecture, with a constant learning rate (0.01), SGD with momentum (0.9) and weight decay (5e-4), batch size of 128, and standard data augmentation (crop and flip). We used 300 training epochs, but found that the baselines that estimate shifts/labels, ELR (Liu et al., 2020), SOP (Liu et al., 2022), NAL (Lu et al., 2022), and ours, benefited from more training epochs and were trained for twice as long. Training for longer might be an advantage with noise-free training, but with label noise it is typically more challenging, as there is more time to overfit.

**Clothing1M and WebVision** Due to Clothing1M and WebVision being much larger-scale datasets, we resort to only training our method. For WebVision, we train on the commonly used subset called mini WebVision (Jiang et al., 2018). We use the InceptionResNetV2 architecture with the same training setup as for CIFAR, except we use a learning rate, weight decay, and batch size of 0.02, 2e-3, 64, respectively. $\alpha$ of 0.999. Also, we found that a temperature scaling of 0.3 and a linear warm-up of the learning rate (first epoch) improved the results. On Clothing1M, we use the standard approach of using a ResNet-50 network initialized with weights from pre-training on ImageNet. We use a learning rate, weight decay, and batch size of 0.001, 0.01, and 32, respectively. $\alpha$ of 0.7. Trained for 10 epochs, where the learning rate was scaled at epoch 5 by a factor of 0.1. Standard data augmentation (crop and flip). In each epoch, a 265664 class balanced subset of the dataset is sampled, as in the work of Li et al. (2020a).

The EMA update rate is set to 0.9999 in all experiments.

## A.3 HYPERPARAMETERS

We choose the range of values to search over is based on the original papers. For HET, we search for temperatures in $[0.1, 0.5, 1.0, 10.0, 20.0]$, and over the number of factors $R$ of the covariance matrix in $[1, 2, 4]$. LN searches over temperatures and $\lambda$s in $[0.5, 1.0]$. Our method searches over the $\Delta$ weighting parameter $\alpha$ in $[0.991, 0.993, 0.995, 0.997, 0.999]$. We treat the label smoothing parameter $\gamma$ for LN as fixed, and set it to 0.001 in all experiments on CIFAR, and 0.1 and 0.01 on Clothing1M and WebVision, respectively. For GCE, we search over $q$ in $[0.1, 0.3, 0.5, 0.7, 0.9]$. For NAN, the search is over $\sigma$ in $[0.1, 0.2, 0.5, 0.75, 1.0]$. For LS, we search for values in $[0.1, 0.3, 0.5, 0.7, 0.9]$. For ELR, we search over $\lambda$ in $[1, 3, 7]$ and $\beta$ in $[0.7, 0.9, 0.99]$. For SOP, we search over learning rates for $u$ in $[0.1, 1, 10]$ and $v$ in $[1, 10, 100]$. For NAL, we search for $\lambda$ in $[0.5, 10, 50]$ and $\beta$ in $[0.97, 0.98, 0.99]$.

All searches are done with a single seed, and the hyperparameters with the highest noisy validation accuracy at the end of training are used to train four more networks with different seeds. The hyperparameters used for different noise rates, noise types, and datasets are shown in Tables 5, and 6.

Table 5: **Hyperparameters used for Synthetic Noise on CIFAR datasets.** For HET we report the temperature and the number of factors ($[\tau, R]$), for GCE $q$, for NAN $\sigma$, for LS $t$, and for LN the temperature ($\tau$) and $\lambda$. ELR $[\lambda, \beta]$, SOP [lr u, lr v], NAL [$\lambda, \alpha$], SGN $\alpha$.

| Dataset | Method | No Noise | Symmetric Noise Rate | | | Asymmetric Noise Rate | | |
|---|---|---|---|---|---|---|---|---|
| | | 0% | 20% | 40% | 60% | 20% | 30% | 40% |
| | HET | [0.5, 4] | [20, 4] | [20, 1] | [10, 2] | [20, 2] | [0.1, 2] | [0.5, 2] |
| CIFAR-10 | NAN | 0.2 | 0.5 | 0.75 | 0.75 | 0.5 | 0.1 | 0.2 |
| | GCE | 0.1 | 0.9 | 0.9 | 0.9 | 0.9 | 0.9 | 0.1 |
| | LS | 0.7 | 0.9 | 0.9 | 0.9 | 0.5 | 0.3 | 0.9 |
| | LN | [0.1, 0.5] | [0.5, 0.5] | [1.0, 0.5] | [1.0, 0.5] | [0.5, 0.5] | [0.5, 0.5] | [0.5, 0.5] |
| | ELR | [1, 0.9] | [3, 0.7] | [7, 0.9] | [7, 0.9] | [3, 0.7] | [7, 0.99] | [7, 0.99] |
| | SOP | [0.1, 100] | [10, 1] | [10, 10] | [10, 10] | [1, 100] | [1, 100] | [10, 10] |
| | NAL | [10, 0.99] | [10, 0.99] | [10, 0.98] | [10, 0.98] | [50, 0.99] | [10, 0.99] | [50, 0.99] |
| | SGN (Ours) | 0.999 | 0.997 | 0.991 | 0.991 | 0.997 | 0.997 | 0.997 |
| | HET | [0.5, 4] | [10, 4] | [10, 4] | [10, 1] | [20, 2] | [20, 1] | [20, 1] |
| CIFAR-100 | NAN | 0.1 | 0.2 | 0.2 | 0.2 | 0.1 | 0.2 | 0.1 |
| | GCE | 0.5 | 0.5 | 0.5 | 0.5 | 0.7 | 0.7 | 0.5 |
| | LS | 0.1 | 0.9 | 0.7 | 0.7 | 0.9 | 0.7 | 0.9 |
| | LN | [0.1, 0.5] | [1.0, 0.5] | [1.0, 0.5] | [1.0, 0.5] | [0.5, 0.5] | [0.5, 1.0] | [0.5, 1.0] |
| | ELR | [7, 0.9] | [7, 0.9] | [7, 0.9] | [7, 0.9] | [7, 0.9] | [7, 0.9] | [7, 0.9] |
| | SOP | [10, 10] | [1, 10] | [1, 10] | [1, 10] | [1, 10] | [1, 10] | [1, 10] |
| | NAL | [50, 0.99] | [10, 0.99] | [10, 0.99] | [50, 0.99] | [50, 0.99] | [50, 0.99] | [50, 0.99] |
| | SGN (Ours) | 0.997 | 0.997 | 0.995 | 0.993 | 0.995 | 0.999 | 0.997 |

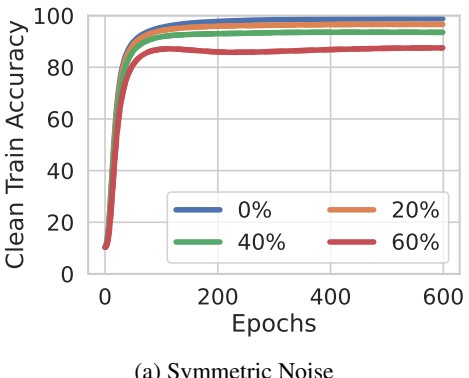 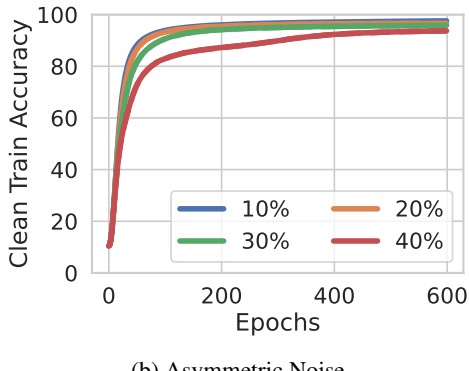

(a) Symmetric Noise          (b) Asymmetric Noise

Figure 3: **Label Correction.** To understand the label correction capabilities of the EMA network, we plot the clean training accuracy during training with various levels of symmetric (a) and asymmetric (b) noise on CIFAR-10.

## A.4   DATASETS

We conduct experiments on the CIFAR-N (Wei et al., 2021b), Clothing1M (Xiao et al., 2015), and (mini) WebVision Li et al. (2017) datasets. The CIFAR-N datasets provide new sets of human labels for the training sets of CIFAR-10 and CIFAR-100, respectively. On the CIFAR-N datasets, we follow the same thorough training setup as for the CIFAR datasets. Clothing1M is a dataset crawled from the web and has a training set of one million images of 14 different types of clothes. The training set has noisy labels due to the automatic assigning of labels based on text information accompanying each image. WebVision v1 is another large-scale dataset with over two million training examples collected by using search engines to query images related to the thousand classes of the ILSVRC 2012 dataset. To speed up experiments, a smaller subset was proposed called mini WebVision (Jiang et al., 2018) that only uses the first 50 classes (of Google subset). Due to the time necessary to conduct experiments on Clothing1M and (mini) WebVision, we forgo our thorough training setup on the CIFAR related datasets, and only train our method. Here, we also compare with two new baselines, Forward (Patrini et al., 2017), and Co-teaching (Han et al., 2018).

## A.5   ADDITIONAL LABEL CORRECTION EXPERIMENTS

In Figure 3, similarly to Figure 2, we plot the clean training accuracy of the EMA network predictions during training when training on various amounts and types of noise on CIFAR-10. We observe a significant improvement in the label correction ability, especially on symmetric noise, compared to the challenging CIFAR-100 dataset.

Table 6: **Hyperparameters used for Natural Noise** For HET we report the temperature and the number of factors ($[\tau, R]$), for GCE $q$, for NAN $\sigma$, for LS $t$, and for LN the temperature ($\tau$) and $\lambda$. ELR $[\lambda, \beta]$, SOP [lr u, lr v], NAL $[\lambda, \alpha]$, SGN $\alpha$.

| Method | CIFAR-10N | | | | | CIFAR-100N |
|---|---|---|---|---|---|---|
| | Aggregate | Random 1 | Random 2 | Random 3 | Worst | |
| HET | [20, 1] | [10, 4] | [0.1, 1] | [20, 1] | [10, 4] | [10, 1] |
| NAN | 0.5 | 0.75 | 0.5 | 0.5 | 0.75 | 0.2 |
| GCE | 0.5 | 0.9 | 0.7 | 0.9 | 0.9 | 0.5 |
| LS | 0.9 | 0.5 | 0.9 | 0.5 | 0.7 | 0.9 |
| LN | [1.0, 1.0] | [0.5, 0.5] | [1.0, 0.5] | [0.5, 0.5] | [0.5, 0.5] | [0.5, 0.5] |
| ELR | [3, 0.7] | [7, 0.99] | [7, 0.9] | [3, 0.7] | [7, 0.99] | [7, 0.9] |
| SOP | [1, 10] | [10, 10] | [10, 10] | [10, 10] | [1, 100] | [1, 10] |
| NAL | [10, 0.99] | [50, 0.99] | [50, 0.99] | [50, 0.99] | [10, 0.99] | [50, 0.99] |
| SGN (Ours) | 0.999 | 0.995 | 0.997 | 0.995 | 0.993 | 0.993 |

## A.6 Connections to Related Works

In this section, we use the notation that $p(y|x)$ and $\tilde{p}(y|x)$ corresponds to the true and noisy label distributions from the underlying generation process, respectively.

ELR (Liu et al., 2020) relies on networks to have not overfitted to noisy labels early in training. They propose to estimate ground-truth labels by averaging per-example predictions of the network during training, which are stored in separate data buffers. In addition to the cross-entropy loss, a term that encourages consistency between the predictions of the network and the estimated labels is used. Due to the memory usage, we believe using buffers to estimate labels do not scale to large datasets and or large number of classes, while our approach of using EMA networks do. Furthermore, the buffers are useless at test time, whereas EMA networks can generalize significantly better than the original network.

**Additive Noise Models**  SOP (Liu et al., 2022) modifies the categorical network prediction by adding a shift followed by a normalization to get the new prediction for a cross-entropy loss. We interpret this approach as the following additive noise model: $\tilde{p}(y|x) = N(p(y|x) + s)$, where $N$ is a normalization function and $s$ is their unconstrained shift stored in per-example buffers. Intriguingly, this resembles our approach of predicting a shift $\Delta$, and can therefore do label correction, but because their addition of the unconstrained shift is done in the simplex, it requires an ad-hoc re-normalization. Furthermore, to effectively learn the shift, gradients of another loss function had to be used, which our method avoids by predicting $\Delta$. As the shift is stored in buffers, it has the same buffer-related issues as ELR.

LN (Englesson et al., 2023) proposes the following noise model: $\tilde{p}(y|x) = S_C(\mu + \epsilon), \epsilon \sim \mathcal{N}(0, \Sigma)$, where $p(y|x) = S_C(\mu)$, and $S_C(\cdot)$ is the softmax centered function. They show how the noise model leads to a Logistic-Normal likelihood (Atchison & Shen, 1980), that has loss reweighting properties. Interestingly, this is a special case of our adapted log-ratio transform approach, corresponding to using the asymmetric alr transform and without modelling $\Delta$, *i.e.*, no label correction.

NAL (Lu et al., 2022) interpolates the one-hot encoded label with the categorical network prediction and use this as the prediction in a cross-entropy loss, where the interpolation parameter is also outputted by the network. Clearly, the influence of noisy examples can be reduced by using an interpolation weight that results in the prediction being close to the label. We interpret this as the following additive noise model: $\tilde{p}(y|x) = (1 - \gamma)p(y|x) + \gamma\delta_y = p(y|x) + \gamma(\delta_y - p(y|x))$, where $\delta_y - p(y|x)$ resembles our $\Delta$, but in the simplex. Notably, unlike SOP, no normalization is needed as the interpolation stays in the simplex. Additionally, they combine their method with an independent label correction approach. Ground-truth labels are estimated similarly to ELR, and therefore this method has the same buffer-related limitations.

**Loss Reweighting or Label Correction Methods**  CMW-Net (Shu et al., 2023) is a loss reweighting method that uses meta-learning to learn the weights, similar to the work of Shu et al. (2019). The main difference is CMW-Net also incorporates extra (task/class) information, leading to more flexibility in learning weighting schemes. Interestingly, Shu et al. (2023) also enhance CMW-Net with label correction (denoted CMW-Net-SL). Instead of learning a loss weight, CMW-Net-SL learns an interpolation parameter between the original one-hot label and a soft label estimate (Equation 14 can be rewritten to the form in Equation 13, in the same way as Equation 12 was), and the new interpolated label is then used in a standard CE loss (no loss reweighting). Similar to NAL, we interpret this as the following additive noise model: $\tilde{p}(y|x) = (1 - v)p(y|x) + v\delta_y = p(y|x) + v(\delta_y - p(y|x))$, where $\delta_y - p(y|x)$ resembles our $\Delta$. Therefore, although the proposed approaches are interesting and novel, we argue neither CMW-Net (loss reweighting) nor CMW-Net-SL (label correction) unifies loss reweighting and label correction.

## A.7 Discussion: A Case for Classification as Regression

Our goal in this section is to make a strong case for why transforming classification to regression is an interesting and potentially promising new research direction.

**Probabilistic Classifiers and Regression**  The goal of classification is to correctly predict which class unseen examples belong to. There are non-probabilistic classification algorithms that directly

predict class IDs, e.g., decision trees (Kotsiantis, 2013), nearest neighbours (Cunningham & Delany, 2021), etc. However, in probabilistic classification, the labels (one-hot encodings) and predictions lives in a continuous multidimensional real space, *i.e.*, the probability simplex. Thus, standard softmax classifiers are essentially regressors. In this work, we just acknowledge this fact and exploit it to map the constrained real space (simplex) to an equivalent unconstrained real space by adapting the well-founded log-ratio transform approach. We are excited about this, as then the entire regression literature is at our disposal. To be clear, viewing classification as regression does not by default solve all our problems or necessarily perform better, but it opens up a new and potentially fruitful research direction to explore.

**A Promising Example**   To show the promise of this direction, we make some simple (shifted Gaussian) noise assumptions in this unconstrained real space that naturally leads to our loss, which we show have properties similar to well-established robustness techniques developed for classification: loss reweighting and label correction. In contrast to classification approaches that combine separate loss reweighting and label correction methods, our approach is simple to understand and analyse from a probabilistic perspective, which is directly due to the regression view, as it opens up the ability to use the Gaussian likelihood.

**Stronger Robustness**   Although our simple Gaussian noise model achieved strong results compared to many recent baselines, we have reason to believe even stronger robustness is possible. For example, we are interested in exploring the subfield of robust statistics (Huber, 2004; Maronna et al., 2019), that have developed regression methods specifically for robustness against outliers and label noise.

**Uncertainty Estimation**   We believe viewing classification as regression is not only exciting from a robustness to label noise perspective, but also from an uncertainty perspective. A common/standard prior in Bayesian statistics is the Gaussian, which is problematic as it is not a conjugate prior for the standard categorical likelihood in classification, leading to the use of approximate inference techniques (Zhang et al., 2018; Nickisch & Rasmussen, 2008). By viewing classification as regression, we get access to the Gaussian likelihood with which all relevant integrals are Gaussian and can be evaluated analytically, see *e.g.*, Gaussian processes regression (Rasmussen et al., 2006). This could lead to more efficient algorithms with better noise robustness (related to aleatoric uncertainty) and (epistemic) uncertainty estimates for classification. We are excited to explore this in future work.

**Deep Learning Theory**   In the theoretical literature on deep learning, taking a regression perspective of classification can in many cases ease the theoretical analysis. For example, the law of robustness (Bubeck & Sellke, 2021), the neural tangent kernel (Jacot et al., 2018), and the bias-variance tradeoff (Geman et al., 1992), etc. We hope our approach to transform classification to regression could help connect these theories to classification in a principled way.

**Squared Loss for Classification**   Recently, there has been a growing amount of evidence for the practical usefulness of using regression losses for classification. In particular, the squared error loss, which correspond to a Gaussian likelihood with an identity covariance matrix applied on the one-hot encoded versions of the labels. For example, Hui & Belkin (2020) compared using the squared error regression loss with the standard cross-entropy (CE) loss on several classification tasks. They concluded: "Our empirical results suggest amending best practices of deep learning to include training with square loss for classification problems on equal footing with cross-entropy or even as a preferred option.". Another example is the work of Hu et al. (2022) that theoretically and empirically studied the squared loss for training deep learning classifiers. They found several good properties: fast convergence rates, strong adversarial robustness, and low calibration error. We believe our adapted log-ratio transform approach is a sound alternative to directly regressing on the one-hot labels, as the regression predictions can naturally be mapped back to the simplex via the inverse transform.

We believe our work provides a promising direction towards robust deep learning in the presence of label noise, and we are excited to further improve robustness by exploring the vast (robust) regression literature.

## A.8 Evolution of Validation Accuracy

To investigate any potential overfitting problem, we study the clean validation accuracy during training for the EMA network for various symmetric and asymmetric noise rates on CIFAR-10 and CIFAR-100, see Figure 4. We find there are no signs of overfitting. In fact, based on the training curves, the performance of SGN could potentially be improved further if training for longer on the higher noise rates.

## A.9 Additional Ablation Study

In Table 1, we found that SGN outperforms CE when no noise is added. To better understand the reason for this, we perform an ablation study, similar to Table 4, where we study the performance of SGN when we systematically deactivate its components: loss reweighting, label correction, and EMA predictions. The results can be found in Table 7. We find that the main reason for the improvement over CE is due to the EMA predictions. As there is no added label noise, and loss reweighting and label correction are directly related to modelling label noise, they do not bring the same strong improvements as was observed for label noise in Table 4.

## A.10 Hyperparameter Sensitivity on Generalization

To investigate the sensitivity of our method to the only hyperparameter $\alpha$ that the gradual introduction of the shift $\Delta$, we plot the clean validation accuracy on CIFAR-100 for various settings of $\alpha$ of the EMA network, see Figure 5. We find $\alpha = 0.995$ performs well in all setups, and that $\alpha = 1$ (no shift) performs much worse, showing the importance of the shift on robustness.

## A.11 Symmetric and Asymmetric Noise

Symmetric noise of X% means that there is an X% risk for each example that their label is replaced by a uniformly random new label. Similarly, for asymmetric noise, there is a risk the label is replaced, but now to a perceptually similar class, e.g., images with cat labels are replaced with a dog label.

In the symmetric case, even after applying noise with a noise rate above 50%, out of all examples with a particular label, the majority of them will still be correctly labelled. An example will make this clearer, consider the case with three classes, A, B and C, with 50% noise (and for simplicity, the randomly selected new label cannot be the original label). Then, after the noise process, out of the examples that originally had a label of A, 50% are still As, 25% now are Bs, and 25% are Cs. With a similar argument for other classes, we find that out of all examples with a label of A after the noise process, 50% of the examples were originally As, 25% Bs, 25% Cs. In fact, as long as the noise rate is below (K-1)/K, the true class will still be in majority after the noise process, so there is still a signal to learn from.

The asymmetric noise case is different. As some classes are only changed to a particular other class (e.g., cat to dog), for the true class to still be in majority after the noise process, the noise rate must be below 50%.

To summarize, the noise rates must be less than $\frac{K-1}{K}$ and $\frac{1}{2}$ for symmetric and asymmetric noise, respectively. This makes it possible to have larger than 50% symmetric noise rates on datasets such as the CIFAR datasets ($K = 10$ and $K = 100$).

## A.12 Comparison with SOTA Methods

We argue that a direct comparison between SGN and state-of-the-art (SOTA) methods like ELR+ (Liu et al., 2020) and DivideMix (Li et al., 2020a) is unfair. This is because these methods have several add-ons only to improve performance, *e.g.*, training two networks at the same time, averaging the predictions of the two networks at test time, and using a stronger data augmentation technique called mixup (Zhang et al., 2017b). To show the importance of these add-ons and the effectiveness of our approach, we propose to keep our original training setup exactly the same and only use a single add-on in the form of mixup (mixup alpha of 0.8). Despite only using a single add-on, SGN with mixup achieves higher top-1 accuracy on WebVision compared to both DivideMix and ELR+ (baseline results are gathered from Liu et al. (2020)), see Table 8.

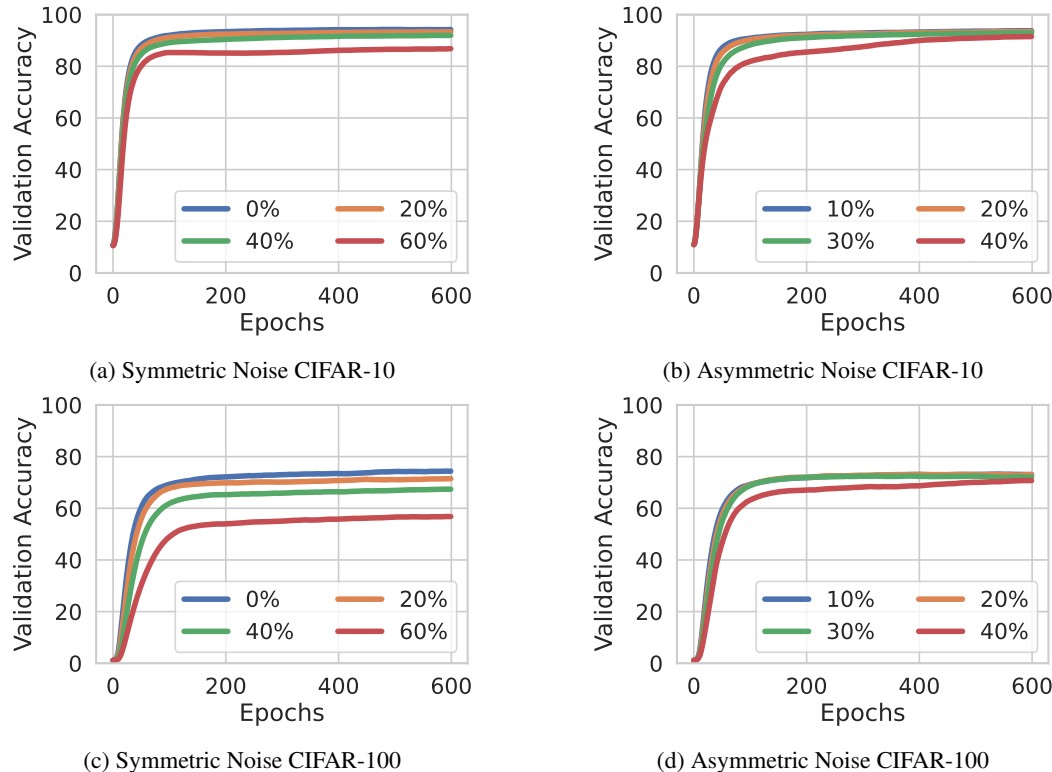

Figure 4: **Generalization.** We plot the clean validation accuracy during training with various levels of symmetric (a, c) and asymmetric (b, d) noise on CIFAR-10 (a,b) and CIFAR-100 (c, d) for the EMA network. There are no signs of overfitting.

Table 7: **Ablation Study.** To better understand why SGN outperforms CE on the CIFAR datasets without added label noise, we perform an ablation study where we systematically deactivate the three components of the method: loss reweighting (LR), label correction (LC), and EMA prediction (EP). The test accuracy for CE without noise is $90.67 \pm 0.80$ and $64.87 \pm 0.88$ for CIFAR-10 and CIFAR-100, respectively. In this setting, we find that the main reason SGN outperforms CE is due to EP.

| LR | LC | EP | CIFAR-10 | CIFAR-100 |
|---|---|---|---|---|
| ✗ | ✗ | ✗ | $90.34 \pm 0.44$ | $66.63 \pm 1.14$ |
| ✓ | ✗ | ✗ | $88.75 \pm 1.08$ | $62.27 \pm 0.72$ |
| ✓ | ✓ | ✗ | $89.09 \pm 1.02$ | $63.44 \pm 1.72$ |
| ✗ | ✗ | ✓ | $\mathbf{94.75 \pm 0.08}$ | $\mathbf{73.96 \pm 0.12}$ |
| ✓ | ✗ | ✓ | $94.07 \pm 0.12$ | $72.42 \pm 0.44$ |
| ✓ | ✓ | ✓ | $94.12 \pm 0.22$ | $\mathbf{73.88 \pm 0.34}$ |

Table 8: **Natural Noise: (mini) WebVision.** We show that simply adding strong augmentations (mixup) to SGN greatly improves the performance on WebVision, making it competitive with more complex state-of-the-art methods.

| Method | WebVision Validation | | WebVision Test (ILSVRC12) | |
|---|---|---|---|---|
| | Top 1 | Top 5 | Top 1 | Top 5 |
| ELR+ (uses mixup) | 77.78 | 91.68 | 70.29 | 89.76 |
| DivideMix (uses mixup) | 77.32 | 91.64 | 75.20 | 90.84 |
| SGN w/o mixup | $76.12 \pm 0.36$ | $90.74 \pm 0.29$ | $72.72 \pm 0.17$ | $90.35 \pm 0.28$ |
| SGN w/ mixup | $78.21 \pm 0.18$ | $90.82 \pm 0.31$ | $75.53 \pm 0.47$ | $89.95 \pm 0.21$ |

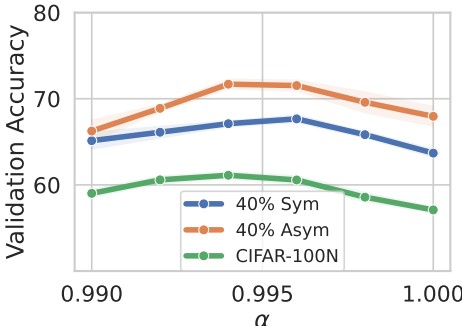

Figure 5: **Hyperparameter Sensitivity.** We plot the clean validation accuracy during training for various values for the hyperparameter $\alpha$ of the EMA network on CIFAR-100 with 40% symmetric and asymmetric noise, and CIFAR-100N. We find $0.995$ is a robust choice for $\alpha$ that performs well in all setups, and that $\alpha = 1$ (no shift) performs worse, showing the robustness improvements of our label correction procedure.

For SGN with mixup, the validation accuracy continually improved, so we report the accuracies at the end of training rather than relying on early stopping. These results are encouraging, and we see no reason why more add-ons would not improve the performance further.

