# OpenReview forum: "Robust Classification via Regression for Learning with Noisy Labels"
_ICLR.cc/2024/Conference — ICLR 2024 poster_

### Official Review · Reviewer_prcP · 2023-10-18

**Soundness:** 3 good
**Presentation:** 3 good
**Contribution:** 2 fair
**Rating:** 5
**Confidence:** 4

**Summary:**

This paper proposes an approach for the task of classification with noisy labeled training data.
The proposed method first uses a bijective function $f$ to map the discrete labels into a continuous label space.
With continuous labels, the method can be naturally formulated as a regression problem, the loss reweighting and label correction techniques are used for the regression task to uncover the true continuous label.
Lastly, the inverse of bijective function $f$ is used to transform the predicted true label to true discrete label.
Empirical experiments are conducted to show the effectiveness of the proposed method in comparison to existing approaches.

**Strengths:**

- The paper's presentation of the proposed method is straightforward and easy to follow.

- The empirical experiments are relatively comprehensive.

**Weaknesses:**

- The primary limitation of the paper is the absence of a clear rationale behind the decision to transform the classification problem into a regression problem. The paper would significantly benefit from a more in-depth analysis of why this transformation is considered a "better" approach for handling noisy labels. Currently, the proposed method appears to be a compilation of established techniques, and addressing this aspect could enhance the quality and impact of the paper.

- The reviewer has a general impression that some key details are missing in the description of the method and the experiments, the reviewer will point them out below.

- Method description
    - P4: The use of an equal weight vector $u$ in (4) for label smoothing should be clarified. What would be the effect of using a vector of all 1s scaled by other factors? Is this equivalent to adjust $\gamma"? The authors are encouraged to provide further clarification on this matter.

    - P4: Regarding loss reweighting via a Gaussian noise model, there may be a computational challenge with evaluating the density function of a multivariate normal due to the inversion of covariance matrices, especially in high-dimensional cases. It appears that the matrix inversion lemma might have been employed as a workaround. Can the authors please confirm and elaborate on this technique? Additionally, it is not entirely clear how $\mu_{\theta}$ and $r_{\theta}$ correspond to the output of the neural network. Are they concatenated as a long vector, or are they separate outputs with distinct final linear layers?

- Experiments:
    - In Table 1, it is unclear why the symmetric and asymmetric noise rates are different. Moreover, it does not make sense to the reviewer that a noise rate is more than 50%, please clarify.

    - While Table 1 presents interesting results, there remains a degree of skepticism regarding its persuasiveness. As the authors pointed out that "SGN significantly outperforms CE when there is no added label noise." This prompts a reasonable inquiry into whether the relatively good performance of the proposed method is contingent on the suboptimal performance of the baselines. The authors posit that the EMA framework may contribute to this observed improvement. Consequently, the reviewer suggests conducting an additional experiment similar to the one presented in Table 4 under a noise-free setting to validate this conjecture and bolster the credibility of the results. Such an addition would further fortify the paper's empirical foundation.

    - The description of the experiments could benefit from more clarity. While the reviewer understands there is a space limit for conference paper, providing essential details in the appendix, especially for a paper predominantly reliant on empirical results, is advisable. For instance:

        - A brief overview of the datasets would greatly assist readers who are new to the field of noisy label classification.

        - The authors should point out aggregate, random, worst corresponding to three sets of noisy labels in Table 2. Otherwise, the table is difficult to understand for people who does not know the dataset well.

- Minor issues:
    - P2: The last two sentences in the "compositional data" paragraph lack clarity regarding what "several problems" and "this problem" refer to.
    - P2: The final sentence in the "transforms" paragraph could be more informative. Instead of stating that "several solutions exist," the authors could refer to the specific technique employed later in the paper for a more insightful description.
    - The expression in (2) on P3 is statistically incorrect. The conditional distribution of the transformed continuous label given features is a normal distribution rather than the mariginal distribution of the label itself.
    - P5: The first sentence "we describe our experimental setup of our experiments" in Section 4, appears redundant.
    - The LaTeX formatting for quotes, as exemplified by "worst" on P7, may require adjustment.

**Questions:**

Please refer to the questions in the previous section.

---

> ### Author Response · Authors · 2023-11-16
> **Thank you for your feedback. [1/2]**
>
> Thank you for the time and effort you put into the detailed review of our work. Our goal with this work is to introduce an adaptation of the well-founded log-ratio transform approach to the classification with noisy labels literature, and show the promise of the direction by devising a method unifying the ideas of loss reweighting and label correction from classification. We are glad you found the explanation of our method to be straightforward and easy to follow.
>
> **Rationale for transforming classification to regression**
>
> We agree that the motivation for transforming classification to regression could be improved, which is why we have now provided a detailed discussion on this in Section A.7. We do not claim this view by default is a “better” approach for handling noisy labels in classification. However, we do claim the regression view that our adapted log-ratio transform approach provides, opens up a new research direction to leverage sound and time-tested methods from the (robust) regression literature. Please also see the “Clarification on our contributions and novelty” paragraph in the response to Reviewer U8oG.
>
> **Clarification of label smoothing**
>
> As the log-ratio transform is only defined for categorical distributions inside the probability simplex (no component can be zero), we need a way to transform class IDs from classification to the interior probability simplex. We achieve this by using the simple and common regularization technique called label smoothing [1], and we provide the original definition in Equation 4 (see Section 7 in the original paper). The idea is to smooth the original one-hot encoded version of the label by interpolating it with the uniform distribution over K classes (all components equal to 1/K).
>
> The reviewer wants a clarification if it is possible to replace the uniform distribution with a vector of ones scaled by other factors than 1/K. An important property that is exploited in label smoothing is that interpolating two categorical distributions is another categorical distribution, for interpolation weights in $\[0,1\]$. So to clarify if we can use other scale factors, the answer is in general no, as a vector of ones can only be scaled by 1/K to be a valid categorical distribution (sum of components are one). However, it is possible to generalize label smoothing by replacing the uniform distribution with any other categorical distribution. For our purposes, the uniform distribution is a good choice though, as we require the interpolated distribution to have non-zero probability in all components and, a priori, we have no way of justifying putting more probability in any particular class.
>
> **Efficient evaluation of loss function**
>
> The reviewer brings up an interesting question about how the negative log-likelihood of the Gaussian distribution can be efficiently evaluated. In short, to evaluate the loss function, inverses, and determinants of the covariance/scale matrices have to be calculated, which for general matrices are computationally heavy. Indeed, the parameterization we use makes inverse and determinant computations efficient through the Woodbury matrix identity, and the matrix determinant lemma, respectively. This is a common parameterization for the covariance/scale matrix of Gaussian distributions, that is implemented in probabilistic frameworks such as Tensorflow Probability. We agree that a motivation for the particular covariance/scale matrix parameterization was missing, which we have now added to Section A.1.
>
> Regarding $\mu_\theta$ and $r_\theta$, we use a standard neural network (WideResNet 28-2, ResNet 50, or InceptionResNetV2) and replace the last linear layer with two linear layers (heads), one for predicting $\mu$ and one for predicting the rank-one factor $r$ of the scale matrix.

---

> > ### Author Response · Authors · 2023-11-16
> > **Thank you for your feedback. [2/2]**
> >
> > **Clarifications on synthetic noises**
> >
> > The reviewer wonders why the noise rates are different between the symmetric and asymmetric noise types, and how the noise rate can be higher than 50% in the symmetric case.
> >
> > To clarify this, we need to describe the different noise types. Symmetric noise of X% means that there is a X% risk for each example that their label is replaced by a uniformly random new label. Similarly, for asymmetric noise, there is a risk the label is replaced, but now to a perceptually similar class, e.g., images with cat labels are replaced with a dog label.
> >
> > In the symmetric case, even after applying noise with a noise rate above 50%, out of all examples with a particular label, the majority of them will still be correctly labelled. An example will make this clearer, consider the case with three classes, A, B and C, with 50% noise. Then, out of the examples that originally had a label of A, after the noise process, 50% are still As, 25% now are Bs, and 25% are Cs. With a similar argument for other classes, we find that out of all examples with a label of A after the noise process, 50% of the examples were originally As, 25% Bs, 25% Cs. In fact, as long as the noise rate is below (K-1)/K, the true class will still be in majority after the noise process, so there is still a signal to learn from.
> >
> > The asymmetric noise case is different. As some classes are only changed to a particular other class (e.g., cat to dog), for the true class to still be in majority after the noise process, the noise rate must be below 50%.
> >
> > We hope the explanations of the different noise types above makes it clear that methods can be evaluated on higher symmetric noise rates (< (K-1)/K) than the asymmetric noise (< 50%).
> >
> >
> > **Why does SGN outperform CE when there is no added label noise?**
> >
> > Thanks for pointing out our intuitive argument for why SGN outperforms CE in the no noise case. Indeed, we should have performed an experiment to have concrete evidence instead. We have now done this ablation and added it in Section A.9. The experimental results indicate that EMA predictions are the main reason for SGN outperforming CE when no label noise is added. With this additional experiment, we hope the reviewer is convinced that the baselines do not have suboptimal performance. In fact, we have gone to great lengths to achieve the best possible results for the baselines. In the literature, it is common to directly use the optimal hyperparameters reported in the original paper, but we do a search over hyperparameter values (including the optimal) to find what works best in our training setup.
> >
> >
> > **Clarification on the datasets used in experiments**
> >
> > The reviewer suggests adding a brief overview of the datasets in the appendix to assist readers that are new to the noisy labels field. We wholeheartedly agree, and in fact, we provided such a section in Section A.4, which was referred to in the first paragraph of the experiments section (Section 4). As suggested by the reviewer, we will clarify that, e.g., “Random 1” and “Worst” correspond to different label sets of CIFAR-10N to reduce confusion for readers unfamiliar with the dataset.
> >
> > **Minor comments**
> >
> > We appreciate the minor comments, we will incorporate them in the final version of the paper. Thank you!
> >
> > **Final Remarks**
> >
> > We would like to thank you again for your detailed review, and appreciate the feedback on how to improve the clarity of the paper for readers that are new to the noisy labels field and deep learning methods like label smoothing. We hope the clarifications regarding the synthetic noise types, how to efficiently evaluate the loss, the uniform distribution in label smoothing, and the investigation into SGNs performance on no noise, all help to improve the understanding of our paper.
> >
> > According to the initial review, the primary limitation of our work was a lack of motivation for converting classification into a regression problem. As we have now provided a detailed motivation for this in Section A.7, we hope the reviewer will recommend our work to be accepted. We look forward to discussing any remaining concerns you may have.
> >
> > **References**
> >
> > [1] Szegedy C, Vanhoucke V, Ioffe S, Shlens J, Wojna Z. Rethinking the inception architecture for computer vision.

---

> > > ### Comment · Reviewer_prcP · 2023-11-22
> > >
> > > Thank you for the rebuttal, most of my concerns are addressed. However, I would like to highlight that my comments regarding the necessity for a clearer description of some key components within the manuscript have not been addressed in the revised version. I am considering increasing my rating to 6, contingent upon the satisfactory resolution of the aforementioned concerns in the manuscript's writing.

---

> > > > ### Author Response · Authors · 2023-11-22
> > > >
> > > > We greatly appreciate your active participation during the discussion period. We are happy to hear that the rebuttal addressed most of the concerns, and that the reviewer considers increasing their score if the paper is improved according to their feedback. Below, we describe how our revised submission incorporates the feedback:
> > > >
> > > > _Method description_
> > > >
> > > > 1. Above Equation 4, we now make it clearer that we use the standard label smoothing technique.
> > > > 2. In Section A.1, we now have a discussion on how to efficiently evaluate the loss, and that we use two distinct linear layers to predict the mean and variance.
> > > >
> > > > _Experiments_
> > > >
> > > > 1. In Section A.11, we now have details about the symmetric and asymmetric noise types.
> > > > 2. In Section A.9, we now perform an ablation study to better understand why SGN outperforms CE when there is no added label noise, and discuss the results.
> > > > 3. In Appendix A.4, we give a brief overview of all the datasets used in the experiments.
> > > > 4. In Section 4.3’s result paragraph, we now mention that aggregate, random 1-3, and worst are label sets of CIFAR-10N. Due to layout constraints, we were not able to fit this detail into the table caption.
> > > >
> > > > _Minor issues_
> > > >
> > > > 1. We have now reformulated the last sentences in Section 2’s compositional data paragraph.
> > > > 2. The transforms paragraph in Section 2 now has a more informative last sentence, as it introduces our label smoothing solution.
> > > > 3. We have now made the dependence on the input x explicit in Equation 2.
> > > > 4. The first sentence in Section 4 no longer includes “of our experiments”.
> > > > 5. We now adopt proper quotation marks throughout the manuscript.
> > > >
> > > > We believe that these modifications meet your expectations. If any aspect falls short, kindly specify the areas in need of improvement.
> > > >
> > > > Thanks again for your initial review, and response to our rebuttal.

---

### Official Review · Reviewer_U8oG · 2023-10-28

**Soundness:** 3 good
**Presentation:** 3 good
**Contribution:** 2 fair
**Rating:** 5
**Confidence:** 4

**Summary:**

To address overfitting to label noise, this paper proposes to combine two popular methods together, namely loss reweighting and label correction. Specifically, the approach firstly uses isometric log-ratio transform to convert a classification task to a regression task, which is then solved with a shifted Gaussian Noise Model. The loss reweighting is achieved when learning sigma in the Gaussian Noise Model, and the label correction is implemented by adding a shift to the noise model to make up the difference between the noise target and the correct label. Experiments show that the proposed method outperform previous baselines and related works on both synthetically and naturally noisy datasets.

**Strengths:**

1. It is interesting and meaningful to use a (shifted) Gaussian Noise Model to achieve loss reweighting and label correction at the same time, which is very different from previous methods, where the two tasks are considered separately.
2. The experiments show the effectiveness of the method on both CIFAR-10 and CIFAR-100 with both synthetic noise and natural noise.

**Weaknesses:**

1. The contribution is limited or needs refinement. To me, the log-ratio transformation, the Gaussian Noise Model for loss reweighting, and the estimation of the shift are all previous works. The authors should be more explicit on the novelty and contributions.
2. The reason for the choice of the orthonormal basis for the Isometric Log-Ratio Transform is absent. The authors should explain why choosing the Helmert matrix as the basis, and better for analyzing the influence or conducting comparative experiments with different basis.
3. The experiment results on Clothing1M and WebVision dataset with natural noise seem to fail compared with NAL, which though with a different evaluation setup, as Tabel 3 shows.
4. As Figure 2 shows, it takes around 400 epochs for the model to converge. However, figure 2 just depicts the train accuracy, and there can be a potential overfitting problem that the authors should consider.

**Questions:**

1. It is not very clear to me that "The authors propose to have a two-layer neural network with parameters θ to output \mu and \sigma per example" in 3.3. How is the two-layer network trained and how does it work?
2. The method converts the classification task to a regression task using isometric log-ratio transformation. When optimizing, the gradient is computed in the transformed space instead of the origin space. Under this condition, is the optimization still consistent with usual methods where no such transformation is used?

---

> ### Author Response · Authors · 2023-11-16
> **Thank you for your feedback. [1/3]**
>
> We would like to thank you for your time and energy in reviewing our work. In this work, we introduce an adaptation of the well-founded log-ratio transform approach to the literature on classification with noisy labels, and show the promise of the direction by devising a method unifying the ideas of loss reweighting and label correction. We are glad you found this approach interesting, meaningful, and novel. Furthermore, we are glad the reviewer recognizes the effectiveness of our simple method on synthetic and natural noise.
>
> **Clarification on our contributions and novelty**
>
> Thanks for giving us the opportunity to more explicitly describe the novelty and contributions of our work. As the reviewer mentions, our use of a shifted Gaussian noise model for unified loss reweighting and label correction is interesting, meaningful and very different from previous methods. This is a novel contribution, despite building on existing notions such as the log-ratio transform and Gaussian noise models. All novel methods build upon existing mathematical notions.
>
> We introduce and adapt the well-founded log-ratio transform to the modern problem of noisy labels in classification. Our approach makes it possible to view classification as regression, opening up a new research direction to leverage sound and time-tested methods from the (robust) regression literature that spans centuries. One of our contributions is to open up this research direction that we believe has immense potential, as we discuss in the added section in Section A.7. To the best of our knowledge, this is the first time the log-ratio transform has been used for this purpose.
>
> Furthermore, we propose a specific method in this adapted log-ratio transform approach, which is a combination of the ilr transform with a shifted Gaussian noise model, that achieves loss reweighting and label correction in a single simple and unified approach. To the best of our knowledge, this is the first unified method for loss reweighting and label correction for label noise robustness in classification. In Section A.6 (referenced in the baselines paragraph in Section 4.2), we described, in detail, the connections to the most related works. As Reviewer QVE6 found this discussion interesting and valuable for understanding our method in the context of related work, we believe you will too. The closest work to the log-ratio transform approach we could find is the recent work on the loss reweighting using the Logistic-Normal (LN) distribution, which is a distribution over categorical distributions, similar to the Dirichlet distribution. We discuss LN in Section A.6, and even have it as a baseline. In short, LN can be seen as a special case of the more general log-ratio transform approach, corresponding to a particular transform (alr) and a Gaussian likelihood (zero-mean). In contrast, our adapted log-ratio transform approach, makes it possible to use any log-ratio transform (e.g., the ilr transform that does not suffer from the asymmetry issues of the alr transform), and also incorporate any regression technique, e.g., a shifted Gaussian noise model as in SGN, or Gaussian Processes, Student-T distributions, etc. Hence, in addition to the novelty of the adapted log-ratio transform, we propose the SGN method corresponding to the ilr transform and the shifted Gaussian noise model. Furthermore, we theoretically link the shift to changing the label, and propose to exploit this to achieve effects similar to the label correction methods from classification. To the best of our knowledge, this connection between the shift in the Gaussian noise and label correction is novel and one of our contributions.
>
> Another contribution is the practical algorithm for estimating the shift that differs from common label correction methods in classification that use buffers (e.g., ELR, SOP, NAL). In our work, we take a different approach, instead of using buffers, we use EMA networks to directly predict the labels/$\Delta$. Due to the memory requirements, we believe buffers do not scale to large datasets and or large number of classes, while our approach of using EMA networks do. Another limitation with the use of buffers is the storing of a single estimated label per example. This poses challenges, e.g., when using strong augmentation strategies. For example, with CutOut, there is a risk that a crucial part of the image is removed, leading to an unreliable prediction that still affects the estimate of the true label in the buffer. Similarly, the widely-used mixup technique encourages the prediction to have confidence in both of the mixed classes, potentially adding confidence in the wrong class, but the prediction still affects the buffer estimate for the example. This is not an issue of our approach, as the EMA network predicts labels for each augmented input separately. Furthermore, the buffers are useless at test time, whereas EMA networks can generalize significantly better than the original network.

---

> > ### Author Response · Authors · 2023-11-16
> > **Thank you for your feedback. [2/3]**
> >
> > **Optimization in transformed space**
> >
> > We find it unclear what “consistent” and “usual methods” refers to in the last question: “When optimizing, the gradient is computed in the transformed space instead of the origin space. Under this condition, is the optimization still consistent with usual methods where no such transformation is used?”.
> >
> > We assume “consistent” refers to the same class predictions, and that “usual methods” include the standard cross-entropy loss, but we encourage the reviewer to please clarify if we are mistaken.
> >
> > The optimization in terms of gradients are not the same for LN and CE. In fact, that’s one of the main reasons we transform the labels to be able to use the Gaussian likelihood that has a natural way to scale gradients based on the covariance (resulting in the loss reweighting behaviour) and model shifted label noise (resulting in label correction). To simplify the analysis, we consider maximum likelihood estimation. In this case, CE completely overfits to the noisy one-hot labels and therefore confidently predicts all the labels it was given, including noise ones. In contrast, the SGN method can converge to a solution that does not overfit the noisy labels due to the gradient scaling (loss reweighting) delaying overfitting to noisy labels, so that our estimation of the shift of the Gaussian noise model (correcting labels) has enough time to achieve accurate estimates. Therefore, maximum likelihood estimates for CE overfits to the noisy labels, while SGN fits the estimated targets that we have empirically shown is closer to the true targets. Therefore, the optimizations are not consistent, which is a desired property of SGN.
> >
> > **Empirical Performance on Clothing1M and WebVision**
> >
> > We are glad the reviewer recognized the effectiveness of our method on the CIFAR datasets on synthetic and real noise. The reviewer claims SGN “fails” compared to NAL on Clothing1M and WebVision. This is incorrect, as on Clothing1M, SGN (73.9) outperforms/matches NAL (73.6). Furthermore, on WebVision, NAL has a different evaluation setup to all other methods, as mentioned in the text and the caption, and is why their results are marked with a star. All other methods evaluate the top 5 validation and top 1 and 5 test accuracies at the time of highest top 1 _validation_ accuracy. In contrast, NAL keeps track of the highest top 1 and 5 validation and test accuracies ever seen during training separately. That is, all results could come from different epochs (and even different training runs). Although it leads to higher reported results, we believe it is bad practice. With this in mind, the only valid comparison is on top 1 accuracy on the validation set, as there the evaluation setups are the same. There, NAL achieves 77.4, while SGN achieves 77.2. Is “fail” a fair description of SGN having 0.2 percentage points lower validation accuracy (i.e. having 5 fewer samples correctly predicted)?
> >
> > **Clarification on potential issue of overfitting**
> >
> > The reviewer requests an investigation into any potential overfitting problem. To this end, we study the clean validation accuracy during training for various synthetic noise types and rates on CIFAR-10 and CIFAR-100, see Section A.8 in the appendix. Our results conclusively show that there are no signs of overfitting in any of these 16 cases.
> >
> > **Clarification on network architecture**
> >
> > We use a standard neural network (WideResNet 28-2, ResNet 50, and InceptionResNetV2) and replace the last linear layer with two separate linear layers (heads), one for predicting $\mu$ and one for predicting the rank-one factor $r$ of the scale matrix. The parameters of the networks are learnt exactly as in a standard classification with the cross-entropy loss, i.e., with backpropagation and SGD. We will make this clearer in the paper.
> >
> > **Clarification on the choice of basis for the ilr transform**
> >
> > Any orthonormal basis can be chosen for the ilr transform, and we agree with the feedback that our particular choice could have been clearer. The rationale for choosing to extract the basis vectors from a Helmert matrix is that it corresponds to the basis used in the original ilr paper [1], and that it is also the default choice in the statistical R programming language [2]. Furthermore, we found that this basis worked well in practice, and therefore we did not see any reason to look for alternatives. We will clarify this in the paper as well.

---

> > > ### Author Response · Authors · 2023-11-16
> > > **Thank you for your feedback. [3/3]**
> > >
> > > **Final Remarks**
> > >
> > > We would like to thank you again for your review. We believe we have, in detail, addressed all your concerns and questions. We have clarified our contributions and novelty, the network architecture used, and choice of basis for the ilr transform. Furthermore, as suggested by the reviewer, we investigated any potential overfitting problem, and found no evidence to support that.
> > >
> > > It is hard to grasp how the criticism in the review warrants such a low initial rating. However, as we believe our rebuttal addresses all your concerns, we are hopeful that it will lead to a recommendation for its acceptance. We look forward to having an active discussion to address any remaining concerns you may have.
> > >
> > > **References**
> > >
> > > [1] Egozcue JJ, Pawlowsky-Glahn V, Mateu-Figueras G, Barcelo-Vidal C. Isometric logratio transformations for compositional data analysis.
> > >
> > > [2] https://search.r-project.org/CRAN/refmans/compositions/html/ilrBase.html

---

> > > > ### Author Response · Authors · 2023-11-22
> > > >
> > > > Hi Reviewer U8oG,
> > > >
> > > > Thanks for again for your initial review. Unfortunately, soon, it is only 24 hours left of the discussion period. We are eagerly waiting to hear what you think of our rebuttal and if it addressed most of your concerns.
> > > >
> > > > Best regards,
> > > >
> > > > Authors

---

### Official Review · Reviewer_h549 · 2023-10-30

**Soundness:** 3 good
**Presentation:** 3 good
**Contribution:** 3 good
**Rating:** 8
**Confidence:** 4

**Summary:**

This paper first uses log-ratio transformation to convert the classification data set into a regression data set. It treats the regression target as a Gaussian distribution with noise, learns the mean and variance of the Gaussian distribution, and then indirectly weights the loss through the variance. Secondly, this paper gives the above noise a shift and treats the regression target as a Gaussian distribution with offset (non-zero mean) noise to estimate the shift. This method indirectly corrects the labels when converting the regression task into a classification task.

The contributions of this paper include:
(1) It turns loss reweighting and label correction into a unified method instead of being two independent processes.
(2) It uses statistics and regression perspectives to look at the label noise problem in classification tasks, which is relatively novel.

**Strengths:**

- The perspective of the paper is novel, using a regression perspective to perform classification tasks. The unified process of loss reweighting and label correction can also be understood as modeling noise.
- The experiments in the paper are relatively sufficient.

**Weaknesses:**

- The experimental results is lower than SOTA performance, e.g., DivideMix, ELR, etc.
- There exist some unified methods that combines loss reweighting and label correction to enhance robustness against label noise, e.g., CMW-Net [1].
-  What's merit of the regression task. Some strong results and theoretical explanations are necessary.
-  The hyperparameter $\alpha$ is sensitive in Fig.2(c).

[1] Shu J, Yuan X, Meng D, et al. Cmw-net: Learning a class-aware sample weighting mapping for robust deep learning[J]. IEEE Transactions on Pattern Analysis and Machine Intelligence, 2023.

**Questions:**

- $\sigma^2$ of formula 6 should be written as $\sigma_\theta^2$.
- What are the advantages of using regression to solve classification tasks rather than directly solving classification tasks? Would it be too cumbersome to convert the classification problem into a regression problem, and then convert the regression problem into a classification problem?
- The previous methods regard loss reweighting and label correction as two independent processes. So what is the advantage of treating them as a unified method in this paper?

---

> ### Author Response · Authors · 2023-11-16
> **Thank you for your feedback. [1/3]**
>
> We greatly appreciate your time and effort in reviewing our work. We are glad you find our approach of using regression to solve classification tasks to be novel. We are happy the reviewer recognizes that our proposed shifted Gaussian noise model directly leads to a unified method for loss reweighting and label correction, and that the experiments are deemed relatively sufficient. Thanks again, and next we address your concerns and questions.
>
> **State-of-the-art Performance**
>
> Our goal with this work is to introduce an adaptation of the well-founded log-ratio transform approach to the classification with noisy labels literature, and show the promise of the direction by devising a method unifying the ideas of loss reweighting and label correction from classification. Despite its simple design, our method empirically performs well against many strong and recent baselines, such as GCE, ELR, SOP, NAL, etc. The reviewer claims a weakness of our paper is that our method has lower performance compared to SOTA methods, and gives the example of DivideMix and ELR. In fact, we do compare with ELR, and our method achieves as good or better performance on CIFAR-10 and CIFAR-100 with synthetic noise, and on natural noise on the CIFAR-N datasets, Clothing1M, and WebVision, see Tables 1, 2, and 3.
>
> We believe comparing SGN with DivideMix is unfair as it is a significantly more complex method as it combines ideas from sample selection GMMs to identify noisy labels (e.g., [1]), with a semi-supervised learning method [2], with co-training of two networks at the same time (e.g., [3]), and also use stronger data augmentation like mixup. Please see the “Rationale for chosen baselines” paragraph in the response to Reviewer QVE6 for a longer discussion.
>
> We believe achieving strong performance is one important point of comparison, but does not take into account other possible empirical/practical advantages. Although DivideMix achieves excellent empirical performance, which is incredibly important for practitioners, we argue it is hard to fully understand due to all the different components. Our method is simpler in that it follows directly from a Gaussian noise model, leading to an interpretable robustness mechanism in terms of loss reweighting and label correction. Furthermore, DivideMIx was specifically created to be excellent at this particular task, but due to its complexity we believe it can be hard to extend/generalize to other problems. We believe our well-founded probabilistic method is easier to extend, e.g., it can naturally incorporate epistemic uncertainty (e.g., through Monte-Carlo Dropout [4] or Gaussian Processes [5]), making it interesting to both the noisy labels and the uncertainty communities.
>
> Finally, we would like to point out that the reviewing guidelines (https://iclr.cc/Conferences/2024/ReviewerGuide) are clear that a lack of state-of-the-art results does not by itself constitute grounds for rejection, see the FAQ.
>
> **Missing related work**
>
> We performed a thorough literature study to select relevant baselines from the loss reweighting and or label correction categories of methods from top conferences and journals. Unfortunately, we missed this very recent work proposing the meta-learning based CMW-Net method.
>
> CMW-Net is motivated in the context of the exciting research direction into learning loss weights using meta learning. Based on the title, abstract, introduction, and most of the method section, the main novelty seems to extend the meta-learned weight network of MW-Net [6] to also consider extra (task/class) information, leading to more flexibility in learning weighting schemes. However, in the last subsection of the method section (Section 3.6), CMW-Net is enhanced with label correction (denoted CMW-Net-SL). Instead of learning a loss weight, the CMW-Net now learns an interpolation parameter between the original one-hot label and a soft label estimate (Equation 14 can be rewritten to the form in Equation 13, in the same way as Equation 12 was), and the new interpolated label is then used in a standard CE loss (no loss reweighting). Therefore, although the proposed approaches are interesting and novel, we argue neither CMW-Net (loss reweighting) nor CMW-Net-SL (label correction) unifies loss reweighting and label correction. In future work, it would be interesting to explore ideas from CWM-Net-SL to learn $\Delta$ in SGN. Thank you for bringing our attention to this recent related work, which we will describe and reference as there are clear connections.

---

> > ### Author Response · Authors · 2023-11-16
> > **Thank you for your feedback. [2/3]**
> >
> > **Missing related work** (continued)
> >
> > _Side-note_: Confusingly, the CMW-Net paper evaluates on more than 50% asymmetric noise on CIFAR-10 and still achieves above 90% test accuracy. In the standard asymmetric noise on CIFAR-10, there is a risk that a cat is changed to a dog and a dog changed to a cat. However, if the noise rate is above 50% then the cat and dog would have swapped class IDs, and it should therefore not be possible to correctly classify either of these classes. Indeed, the official code contains a bug in the noise implementation [7]:
> >
> > “self.transition = {0:0,2:0,4:7,7:7,1:1,9:1,3:5,5:5,6:6,8:8} # class transition for asymmetric noise”
> >
> > The standard implementation maps cats and dogs to each other, which is not done here. We believe 5 should be mapped to 3, not to itself.
> >
> > **Motivation for transforming classification to regression**
> >
> > We agree that a motivation for transforming classification to regression deserves a thorough discussion, which we have now provided in Section A.7. In short, one of the main reasons is that this opens up a new research direction to leverage sound and time-tested methods from the (robust) regression literature that spans centuries. Thank you for pointing this out, as we believe this improves the motivation of our approach.
> >
> > Note that transforming the classification labels to regression labels can easily be done with a pre-processing step or on the fly during training, similar to label smoothing. At test time, instead of applying softmax to the logits as is done for standard softmax classification networks, we instead apply the ilr transform. This is trivial to implement as it requires minimal changes, and is therefore not cumbersome at all.
> >
> > Our goal with this work is to introduce an adaptation of the well-founded log-ratio transform approach to the classification with noisy labels literature, and show the promise of the direction by devising a method unifying the ideas of loss reweighting and label correction from classification. Theoretically, it is not possible to achieve loss reweighting with the standard categorical likelihood, but our regression approach makes it possible to naturally achieve this with the Gaussian likelihood (higher variance corresponds to a lower weight). Furthermore, the Gaussian likelihood makes it possible to model shifted noise, which we theoretically show is similar to label correction methods in classification. Additionally, in Section A.7, we go over several other advantages of the  regression perspective, see e.g., “Uncertainty Estimation” and “Deep Learning Theory” paragraphs. Finally, we believe our experiments have clearly shown the effectiveness of our regression approach compared to several recent and strong baselines, e.g., ELR, SOP, and NAL.
> >
> >
> > **Hyperparameter sensitivity analysis**
> >
> > Our method has a single momentum hyperparameter $\alpha$ that determines how much of $\Delta$ should be used in the noise model: $t = \mu + (1-\alpha^{e})\Delta$, where $e$ is the current training epoch. In this way, the weight $(1-\alpha^{e})$ is close to zero early in training and gradually goes to 1, causing the labels to transition from the observed to the estimated ground-truth labels during the training process. Note that if $\alpha=1$ then we have no shift in the Gaussian noise model, and the lower the value of $\alpha$ the earlier we make use of the shift $\Delta$ in training.
> >
> > To better understand how $\alpha$ affects the label estimation of the EMA network, we performed an experiment to analyse it empirically. In Figure 2c, we report the training accuracy of the EMA network predictions on the noise-free training labels. Based on these results, the reviewer claims a weakness of our work is that $\alpha$ is sensitive.
> >
> > We want to clarify that it is important that $\alpha$ should change the results, as it shows that the label correction has an important effect on the training. Furthermore, an $\alpha$ of $0.995$ is a robust choice that performs well in all the three cases. We want to clarify that Figure 2c shows the clean training accuracy to relate it to label correction performance, which is different from the generalization of the network to unseen data. In Section A.10 in the appendix, we provide a similar plot to Figure 2c, but evaluated on the clean validation accuracy to see the effect on generalization. Again, $\alpha=0.995$ is a robust choice, and the results clearly show that using no shift ($\alpha=1$) negatively affects generalization, substantiating the importance and robustness of our label correction procedure.

---

> > > ### Author Response · Authors · 2023-11-16
> > > **Thank you for your feedback. [3/3]**
> > >
> > > **Advantage of our unified method compared to combinations of separate methods**
> > >
> > > In general, we believe unified methods are often simpler to implement, understand, analyse, and extend. Furthermore, in practice, combining different components that weren’t designed to work together can cause friction, e.g., due to power struggles between the two components that could require extra balancing hyperparameters.
> > >
> > > In particular, NAL is a loss/gradient reweighting method that is also combined with a separate label correction method to improve performance. As these methods are separate, it is hard to understand the method from a probabilistic perspective. For example, it is unclear how soft labels in classification should be interpreted. How are these related to standard class ID labels? Should one interpret this as changing and having multiple new observed labels? In contrast, our simple method follows directly from a shifted Gaussian noise model, where both the label correction (shift $\Delta$) and the loss reweighting (covariance $\Sigma$) have clear roles. Hence, our approach has a different interpretation completely: instead of changing the observations, our view is strictly related to a label noise model with the original observations intact.
> > >
> > > **Final Remarks**
> > >
> > > We would like to thank you again for your review. We hope our clarifications of our method’s empirical effectiveness, hyperparameters, and motivation all help improve the understanding of our work, and address all your concerns. If not, we are happy to continue discussing the remaining issues. If the concerns are addressed, and given the positive review on novelty, unified method, and sufficiency of experiments, we think it is reasonable to expect the reviewer to recommend our work to be accepted.
> > >
> > > **References**
> > >
> > > [1] Arazo, E., Ortego, D., Albert, P., O’Connor, N. and McGuinness, K., 2019, May. Unsupervised label noise modeling and loss correction.
> > >
> > > [2] Berthelot, D., Carlini, N., Goodfellow, I., Papernot, N., Oliver, A. and Raffel, C.A., 2019. Mixmatch: A holistic approach to semi-supervised learning.
> > >
> > > [3] Han, B., Yao, Q., Yu, X., Niu, G., Xu, M., Hu, W., Tsang, I. and Sugiyama, M., 2018. Co-teaching: Robust training of deep neural networks with extremely noisy labels.
> > >
> > > [4] Kendall A, Gal Y. What uncertainties do we need in bayesian deep learning for computer vision?.
> > >
> > > [5] Rasmussen CE, Williams CK. Gaussian processes for machine learning.
> > >
> > > [6] Shu J, Xie Q, Yi L, Zhao Q, Zhou S, Xu Z, Meng D. Meta-weight-net: Learning an explicit mapping for sample weighting.
> > >
> > > [7] https://github.com/xjtushujun/CMW-Net/blob/main/section4/Feature-independent_Label_Noise/dataloader_cifar.py#L23

---

> > > > ### Comment · Reviewer_h549 · 2023-11-21
> > > > **The concerns are addressed**
> > > >
> > > > Thanks for your detailed response, most of them has addressed the concerns. A confused point is the insight of label correction. The label correction of CMW-Net-SL seems promising. Previous works can be special cases of label correction of CMW-Net-SL.

---

> > > > > ### Author Response · Authors · 2023-11-21
> > > > > **Thank you for your response!**
> > > > >
> > > > > Thank you for actively participating in the discussion. We appreciate it immensely. We are happy your concerns are mostly addressed, and it is seemingly only a confusion about an interpretation of a related work that's remaining, which we clarify next.
> > > > >
> > > > > **Label Correction of CMW-Net-SL**
> > > > >
> > > > > The objective in Equation 14 in the CWM-Net paper is to minimize the following with respect to $\boldsymbol{w}$
> > > > > $$
> > > > > \sum_{i=1}^N [ \mathcal{V}(L_i^{tr}(\boldsymbol{w}), N_i ; \Theta)L_i^{tr}(\boldsymbol{w}) + (1-\mathcal{V}(L_i^{tr}(\boldsymbol{w}), N_i ; \Theta))L_i^{Pse}(\boldsymbol{w}) ]
> > > > > $$
> > > > > For notational simplicity, let $\mathcal{V}\_i = \mathcal{V}(L_i^{tr}(\boldsymbol{w}), N_i ; \Theta)$, and use the cross-entropy loss (which was used in their experiments), i.e., $L_i^{tr} = CE(f_i, y_i) = -y^T_i \log{f_i}$ and $L_i^{Pse} = CE(f_i, z_i) = -z^T_i \log{f_i}$, where $y_i$ and $z_i$ are the one-hot encoded label and the estimated soft label, respectively. Then, we have
> > > > > $$
> > > > > \sum_{i=1}^N [ \mathcal{V}_iCE(f_i, y_i) + (1-\mathcal{V}_i)CE(f_i, z_i)] = \sum\_{i=1}^N [ -\mathcal{V}_i y_i^T \log{f_i} - (1-\mathcal{V}_i) z_i^T \log{f_i} ] = \sum\_{i=1}^N [ -(\mathcal{V}_i y_i^T +  (1-\mathcal{V}_i) z_i^T)\log{f_i} ] = \sum\_{i=1}^N [ CE(f_i,  \mathcal{V}_i y_i +  (1-\mathcal{V}_i) z_i) ]
> > > > > $$
> > > > > Therefore, minimizing the objective in Equation 14 with CE losses, is the same as minimizing a single CE loss with a different (corrected) label, i.e., label correction. However, unlike our SGN method that does loss reweighting (through $\sigma^2\_{\boldsymbol{\theta}}$ in Equation 5) and label correction (through $\Delta$ in Equation 6), CMW-Net-SL with the CE loss only does label correction as the loss does not have the ability to do loss reweighting.
> > > > >
> > > > > The reviewer brings up an interesting point that CMW-Net-SL can be seen as a generalization of other label correction methods. We believe our description above makes this clearer, as e.g., Bootstrapping [1] can be seen as a special case with a fixed meta-network outputting a constant $\mathcal{V}$ for all examples. We have updated our submission to reference the related work of both MW-Net and CMW-Net, and added a description of the CWM-Net (and CMW-Net-SL) in Section A.6.
> > > > >
> > > > > Thanks again for bringing our attention to this recent related work, and for actively participating in the discussion. We hope this response clarifies the insight that the CMW-Net-SL method is doing label correction, but not loss reweighting. As the reviewer mentions that our rebuttal addressed most of their concerns, and hopefully this response clarified the last remaining point, we believe it is helpful that this is reflected in the reviewer’s final review including the rating. We are more than happy to continue the discussion if the reviewer wishes so.
> > > > >
> > > > >
> > > > >
> > > > >
> > > > >
> > > > > **References**
> > > > >
> > > > > [1] Reed S, Lee H, Anguelov D, Szegedy C, Erhan D, Rabinovich A. Training deep neural networks on noisy labels with bootstrapping.

---

> > > > > > ### Comment · Reviewer_h549 · 2023-11-21
> > > > > > **Thanks for your detailed and timely response!**
> > > > > >
> > > > > > All concerns are addressed. I tend to accept this manuscript. I think this manuscript will bring some novel understanding of robust classification.

---

### Official Review · Reviewer_QVE6 · 2023-10-30

**Soundness:** 2 fair
**Presentation:** 3 good
**Contribution:** 3 good
**Rating:** 6
**Confidence:** 3

**Summary:**

The paper presented a method adapted from log-ratio transform approach from compositional data analysis for classification with label noise. The method can be viewed as a combination of two widely studied categories of approaches in the field: loss reweighting and label correction. The authors performed extensive experiments and achieved competitive performance across several synthetic and real-world datasets.

**Strengths:**

The proposed method is a refreshing adaptation of a well-founded statistical technique (log-ratio transform) in a modern research problem in machine learning (classification with label noise). It should inspire a new line of work to the research area while adding a competitive baseline to other future work. The writing is well done and very easy to follow. The contribution is clearly presented and justified by drawing comparison to related work.

**Weaknesses:**

My main criticism is on the paper is in its experimental setup and results. First of all, I absolutely do not believe a paper deserves to be published only if it achieves SOTA results. But unless I missed something important (e.g., did the authors reimplement the method and rerun the numbers perhaps?) it seems that the selected baselines to report in the paper are a very biased set in an attempt to make the method look SOTA. More details in the Questions section below.

**Questions:**

#### Major comments:

The proposed method appears in most result tables as the best-performing method. However, a closer look raises a few questions regarding selection bias in the reported baselines. Taking ELR (Liu 2020) as an example (as I did not go through all the baselines).
(a) At 60% symmetric noise in CIFAR10, Table 1 in the original ELR paper reports 86.12% mean accuracy while Table 1 in the current paper 81.87% which is significantly lower than the original authors' numbers.
(b) The original ELR paper proposes a variant with a few add-ons, ELR+, which can outperform ELR but the current authors did not include. Admittedly such tricks should not be the main contribution of the ELR, but the reported SGN in the current paper also employed a few tricks (e.g., EMA) to improve upon its vanilla version. According to Table 5 of the original ELR paper, by adding some form of weight averaging alone, ELR's vanilla performance can be boosted by a few percentage points too.
(c) There are many other baselines in learning with label noise, such as DivideMix, which can achieve much higher accuracy. For reasons that are not disclosed, such methods are simply not included in the current paper. If comparison to these methods do not make sense or is unfair, disclosing the rationale behind why some methods are selected over others as baselines in the paper still makes sense to me.

#### Minor comments:

Appendix A.6 is a very interesting discussion and adds a lot of value to the understanding of the method in the context of related works. I'd argue that it should be moved to the main text, while some results/tables can be moved to appendix.

---

> ### Author Response · Authors · 2023-11-16
> **Thank you for your feedback. [1/2]**
>
> Thank you for dedicating your time and effort to review our work. We appreciate your thoughtful feedback and the recognition of our approach as a fresh adaptation of the well-established log-ratio transform approach, applied innovatively to classification with label noise. As we carefully wrote our paper, we greatly appreciate the positive comments on our paper being well-written and easy to follow. We are excited that you believe our work should inspire new research directions in the area of label noise robustness in classification. Once again, thank you for your time and support. Next, we address your concerns, questions, and suggestions.
>
> **Training and Evaluation Setup**
>
> Indeed, to have fair and conclusive experiments, we _reimplement all baseline methods_ into a shared code base where we have full control over the training and evaluation setup. This is mentioned in the first sentence in Section 4.1 and in the captions of Tables 1 & 2, but unfortunately the reviewer missed this. We have gone to great lengths to achieve the best possible results for the baselines. For example, in the literature, it is common to directly use the optimal hyperparameters reported in the baselines’ original papers. Instead, for our method and all baselines, we do a method-specific hyperparameter search (that includes the reported optimal HP) to find what works best in our training setup. Furthermore, to have conclusive results, we report the mean and standard deviation of five runs with different random seeds, and do statistical significance tests when comparing results. Due to the shared code base, all methods are equally affected by the random seed, leading to using the same weight initialization, data order in SGD, data augmentation, synthetic noisy labels, etc.
>
> Although we believe this setup leads to fair and conclusive results, training and evaluating all these methods require a lot of computational resources, which leads us to limit our studies to a smaller WideResNet 28-2 on the CIFAR datasets. As all methods use this network, our results are still fair and conclusive, but at the expense of a higher test error. Achieving excellent empirical performance is important for practitioners, but this was not the goal in this work. Our goal is to develop a simple well-founded statistical method that is still easy to implement, understand, extend, debug, and use in practice, and to compare this with existing related methods in an as fair and conclusive way as possible.
>
> **Rationale for chosen baselines**
>
> We agree that justifying the choice of baselines is important, we will add discussion similar to the one below to the final submission. Thanks for letting us clarify our rationale for choosing baselines.
>
> _Methods that build on similar ideas._ We have tried our hardest to find relevant and recent baselines that are related to loss reweighting and label correction from top conferences and journals. For example, in Section A.6, we argue there are clear connections between our work and methods like ELR, SOP, LN, and NAL, that can either be seen as loss reweighting and or label correction methods. We argue this is the most important comparison, as we see how our method currently compares with the methods that have taken a similar approach.
>
> _Other noisy labels methods._ Furthermore, we compare with many strong and commonly used baselines outside the loss reweighting and label correction categories, e.g., GCE from the robust loss functions category, and LS and NAN from the regularizations methods. This is informative to understand how our categories currently compare in terms of performance to other categories of algorithms to improve robustness against label noise. In our view, performing worse than baselines from other categories is not necessarily negative, as the research is in different stages of development, and categories of methods performing poorly today could achieve SOTA performance in the future. Therefore, these comparisons are intended for laying out context, and we believe it is important to consider and spread a diverse set of ideas and not limit the research field to what currently achieves the best performance.
>
> _SOTA methods._ Our goal in this work is to adapt the well-founded log-ratio transform approach to classification and apply it to the problem of label noise robustness by proposing a simple and unified method for loss reweighting and label correction. If our goal instead had been to achieve the current SOTA performance (a different but also important goal), we would have taken a different empirical direction. In general, the current SOTA methods are complex and compute-intensive that typically combine several methods. For example, in our training setup, our method has comparable training time to ELR, while ELR is twice and five times faster to train compared to ELR+ and DivideMix (Table I.1 in the appendix of the ELR paper), respectively.

---

> > ### Author Response · Authors · 2023-11-16
> > **Thank you for your feedback. [2/2]**
> >
> > **Rationale for chosen baselines** (continued)
> >
> > Regarding ELR+, there are several differences between ELR and ELR+: i) using EMA network instead of original network to update the label estimation buffer, ii) training two networks at the same time (target for one network is the prediction of the other), iii) at test time an ensemble of these two networks are used, and iv) using stronger augmentation in the form of mixup. Note, that the use of EMA networks is _one of several add-ons_ to the vanilla ELR method to improve performance. In contrast, the EMA network in SGN is not an add-on, it is an integral part of the _vanilla_ SGN method as it predicts the shift of the Gaussian noise. At test time, we use the EMA network over the original network, but this is not an add-on for better performance, but a choice, as we already have access to the EMA network at no extra cost. To be absolutely clear, there is no vanilla SGN vs SGN+ in our paper, there is only the vanilla SGN. As ELR+ contains several add-ons on top of the vanilla ELR, would you agree that comparing SGN with ELR+ is unfair?
> >
> > Similarly, DivideMix combines ideas from sample selection GMMs to identify noisy labels (e.g., [1]), with a semi-supervised learning method [2], with co-training of two networks at the same time (e.g., [3]), and also use stronger data augmentation like mixup. Again, we believe comparing our shifted Gaussian noise model approach with this complex and resource intensive method is therefore unfair, but we look forward to hearing your opinion.
> >
> > Indeed, to boost the performance of our method, we could also create a complex (and resource intensive) method by combining our approach with several other methods, e.g., using two networks, with stronger augmentation, combine it with MixMatch, ensembling at test time, etc. Although this is almost certainly going to improve performance, is it reasonable for every new label noise method to have to create a complex method to compare with SOTA? Do you agree that these comparisons are, in many cases, largely measuring the effectiveness of the choice of how many and what add-ons were used to create the complex method? We believe that comparing the core idea of methods is more conclusive because there are fewer confounders, which improves our understanding of which ideas warrant further investigation. Does this sound reasonable? We believe comparing core ideas of methods is also fairer to researchers with limited computational resources, but we look forward to more discussion if the reviewer is not convinced.
> >
> > **Issues in directly comparing with results from other papers**
> >
> > You bring up a good point about our results being lower than those reported in the ELR paper. As mentioned above, the main difference in our setups is that they use a larger network (ResNet 34) than us (WideResNet 28-2). To highlight the difference, their results with the notoriously overfitting CE loss on 10% asymmetric noise outperforms our CE results with no added label noise. This makes it clear that directly comparing results with other papers is not valid due to differences in training and evaluation setups. If one is only interested in comparing with results from other papers, our setup is very similar to NAN [4]. Our results with SGN are significantly better than all the reported results in that paper. For example, on 40% asymmetric noise on CIFAR-100, their best test accuracy is 63.90, whereas our SGN achieves 71.01.
> >
> > **Connections to related work**
> >
> > We are excited that you also found our discussion on the connections to related work to be valuable for understanding. Based on your suggestion, we will incorporate this discussion in the main paper.
> >
> >
> > **Final Remarks**
> >
> > We would again like to thank you for the detailed review and support, and for recommending our work to be accepted. We believe that our detailed explanations of the training setup (re-implement and re-run all methods) and the rationale behind our chosen baselines should have hopefully addressed your concerns. Finally, we are pleased that the reviewer saw the contributions and novelty of our work and the new line of research it should inspire, which the other reviewers missed and is therefore their main criticism. We hope our rebuttal contributes to a stronger recommendation for our paper's acceptance.
> >
> > **References**
> >
> > [1] Arazo, E., Ortego, D., Albert, P., O’Connor, N. and McGuinness, K., 2019, May. Unsupervised label noise modeling and loss correction.
> >
> > [2] Berthelot, D., Carlini, N., Goodfellow, I., Papernot, N., Oliver, A. and Raffel, C.A., 2019. Mixmatch: A holistic approach to semi-supervised learning.
> >
> > [3] Han, B., Yao, Q., Yu, X., Niu, G., Xu, M., Hu, W., Tsang, I. and Sugiyama, M., 2018. Co-teaching: Robust training of deep neural networks with extremely noisy labels.
> >
> > [4] Chen P, Chen G, Ye J, Heng PA. Noise against noise: stochastic label noise helps combat inherent label noise.

---

> > > ### Comment · Reviewer_QVE6 · 2023-11-21
> > > **Thanks for the detailed rebuttal.**
> > >
> > > Thank you for the detailed rebuttal. I do think that most of my concerns are addressed, however I'd like to offer just a few paragraphs to the discussion.
> > >
> > > **Re-implementation of other baselines**
> > >
> > > I did notice the reimplementation remark in Sec 4.1 but it is hard to grasp the nuanced differences (e.g., how close the re-implementation matches the original implementation) since it is practically _almost_ impossible to re-implement any other paper. This could add to the impression of a _conflict of interests_ when the current paper proposes a competing method and the baseline numbers in the current paper appear to be _somewhat consistently_ lower than those in the original papers. Having said that, I do think it is incredible that the authors re-implemented the other baselines in a single codebase for as conclusive a comparison as possible. I do hope that the codebase will be published if the paper gets accepted. I also hope our discussion about this can help any future reader clear some of their doubts, if they may fall under the same impression as I did.
> > >
> > > **Training tricks/add-ons**
> > >
> > > Let me try to rephrase where my concerns lie.
> > >
> > > 1. Since the paper is IMHO a practical adoption of well-established statistical techniques, an interested reader may wonder why they should use SGN and not other SOTA. So I agree with the authors that a comparison without any add-ons is crucial, but I do not think it is as clear as the current form of comparison. For instance, the authors had EMA as part of SGN natively, but not part of vanilla ELR due to the fundamental difference in their methodological design. If we know EMA alone can boost ELR by a substantial margin, one may wonder how much the claimed improvement of SGN is due to the EMA part of SGN, despite it is an inherent part of SGN. I do not think it is necessary the authors do such additional experiments, but just to lay out some counter arguments.
> > >
> > > 2. I do believe that keeping a few ablations with add-ons are also important. It is not a showcase of how many add-ons one can find or how those powerful add-ons are standalone. But it helps the readers understand how much improvement they can expect to push if they would like to adopt the proposed method in practice, as well as how extendable the current work is (with codebase released). It also offers a more SOTA comparison since many other papers employ different sets of add-ons to boost the performance.

---

> > > > ### Author Response · Authors · 2023-11-22
> > > > **Thank you for your response! [1/2]**
> > > >
> > > > Thank you for your patience. We greatly appreciate your active participation during the discussion period, and we are pleased to hear that you think most of your concerns have been addressed. Next, we gladly go over the discussion points you raised.
> > > >
> > > > **Response to: Re-implementation of other baselines**
> > > >
> > > > If we understand correctly, the reviewer brings up a conflict of interest in re-implementing baselines, as then implementation errors could reduce the performance of the baselines and therefore make the proposed method perform well in comparison. We agree that this could be a problem, but we are confused as to why the reviewer doubts our particular experimental results. The only provided reason seems to be due to a claim of a lower performance compared to the baseline results in other papers. The reviewer seems to have missed that we pointed out the flaws in directly comparing with results from different training setups (most importantly, different network architectures), in the “Issues in directly comparing with results from other papers” paragraph in our initial rebuttal. To hopefully make an even stronger case for the convincingness of results with re-implemented baselines in general, and our empirical results in particular, we now provide several additional arguments.
> > > >
> > > > _Strong performance when compared with a similar training setup._ We propose to perform statistical significance tests (t-test with a level of significance of 0.05) between our baseline results and the results reported in the NAN paper [1], which has the most similar training setup to ours (same network architecture). We compare the symmetric and asymmetric noise results in our Table 1 for 40% noise with the results in Table 1 and 2 of NAN. If the difference between ours and their results is not statistically significant, we mark it as 0, and if the difference is statistically significant and better or worse, we mark it with +1 or -1 (two-tailed P value in parentheses), respectively.
> > > >
> > > > | Method | C10 Symmetric | C10 Asymmetric | C100 Symmetric | C100 Asymmetric |
> > > > | ------ | ------------- | -------------- | -------------- | --------------- |
> > > > | CE     | +1 (0.0083)            | 0 (0.4126)              | +1 (0.0047)             | 0 (0.1145)              |
> > > > | GCE    | +1 (<0.0001)            | 0  (0.2232)            | +1 (<0.0001)             | 0 (0.9561)              |
> > > > | LS     | \-1 (0.0289)           | +1 (0.0045)            | +1 (0.0005)             | \-1 (<0.0001)             |
> > > > | NAN    | 0 (0.6125)            | \-1 (<0.0001)           | 0 (0.5452)             | \-1 (< 0.0001)             |
> > > >
> > > > Our comparisons show that in 25% of the cases our results are worse, but in 75% of the cases our results are the same (37.5%) or better (37.5%). This conclusively shows that our baseline results are strong when compared with papers with a similar training setup. However, again we believe comparing with the results of other papers is problematic due to differences in training setup and that re-implementing into the same code base results in a better comparison. This quantitative investigation in tandem with the consistently observed improvement of SGN over the re-implemented baselines on CIFAR datasets should make the results conclusive.
> > > >
> > > > _Hyperparameter search for baselines._ We have in fact gone to great lengths to achieve strong performance for the baselines. In the literature, if baselines are re-implemented it is common to directly use the optimal hyperparameters reported in the original paper, but, instead, we do a search over hyperparameter values (including the optimal) to find what works best in our training setup. For fairness, the search space, i.e., size of the possible hyperparameters set, for the baselines is equal (or larger) than that of SGN.
> > > >
> > > > _We also compare with results reported in other papers._ The reviewer’s concern about a conflict of interest is mitigated by the fact that we do not only compare with results from our re-implemented baselines. In fact, for Clothing1M and WebVision, we compare with the results reported in the original papers, mainly due to the computational burden. We believe this combined setup can be an ideal in the context of the reviewer’s question, as we both re-implement on CIFAR, but re-use results on WebVision and Clothing1M and consistently show a good performance for SGN compared to the baselines
> > > >
> > > > _Release of code._ In all our previous published works, we have released the code to reproduce our experiments. We will release the code (including baseline implementations) with this paper as well. We have updated the paper (abstract) with a link to a currently empty anonymous GitHub project, where we will release our code as soon as possible.
> > > >
> > > > _Re-implementing baselines is somewhat common._ We are not the first to re-implement baselines, in fact, it is somewhat common to do so in the noisy labels field, see e.g., [1, 2, 3, 4, 5, 6].

---

> > > > > ### Author Response · Authors · 2023-11-22
> > > > > **Thank you for your response! [2/2]**
> > > > >
> > > > > **Response to: Re-implementation of other baselines** (continued)
> > > > >
> > > > > We hope this discussion removes any doubts the reviewer has about our empirical results. We are happy to hear that the reviewer thinks it is incredible that we re-implemented all the baselines (for the CIFAR experiments) to have conclusive comparisons. Indeed, this requires significantly more time, effort, and computational resources, but we believe it is worth it to have reliable comparisons.
> > > > >
> > > > > We agree with the reviewer that this discussion can be useful for future readers, so we will include a summary of the points in the final version of the main paper.
> > > > >
> > > > > **Response to: Training tricks/add-ons**
> > > > >
> > > > > 1. We interpret the reviewer’s argument to be that it is unclear how much of the improvement of SGN over ELR is due to using an EMA network instead of a buffer. If so, we believe this is a point that requires either changing our SGN method to use buffers, or ELR to use an EMA network instead of a buffer, and perform experiments to explore this. Unfortunately, there is not enough time left of the discussion period to properly perform experiments to look into this, and the reviewer mentioned they deemed such experiments unnecessary anyway. However, we believe the ablation study in Table 4 clearly shows that all components of SGN are important: loss reweighting, label correction and EMA predictions. Furthermore, our latest results suggest SGN has other benefits over just incorporating an EMA network, as SGN with mixup outperforms ELR+ (which includes mixup, EMA networks, and other add-ons) on the WebVision dataset.
> > > > > 2. We thank the reviewer for clarifying their practical perspective on incorporating add-ons on top of vanilla methods. We agree that ablation studies are important in understanding the performance impact of different parts and add-ons of methods. Furthermore, we do see the relevance of incorporating add-ons for practitioners. Having said that, our point is that one has to strike a good balance between catering to practitioners and researchers. We are concerned that the research community may be heading towards overly catering to practitioners and making it virtually a requirement to incorporate add-ons and achieve state-of-the-art performance. We believe there should be room for novel directions that show conclusive promise but not yet necessarily achieve SOTA performance, as exploration is indeed a key aspect of science.
> > > > >
> > > > >
> > > > > **Final remarks**
> > > > >
> > > > > We would like to again thank the reviewer for actively participating in the discussion period. As the reviewer mentions that they think our rebuttal addressed most of their concerns, and hopefully this discussion removed any reasonable doubt about our empirical results with re-implemented baselines, we believe it is helpful that this is reflected in the reviewer’s final review including the rating. If there are any remaining concerns, we are more than happy to discuss them.
> > > > >
> > > > >
> > > > > **References**
> > > > >
> > > > > [1] Chen P, Chen G, Ye J, Heng PA. Noise against noise: stochastic label noise helps combat inherent label noise.
> > > > >
> > > > > [2] Wang Y, Ma X, Chen Z, Luo Y, Yi J, Bailey J. Symmetric cross entropy for robust learning with noisy labels.
> > > > >
> > > > > [3] Ma X, Huang H, Wang Y, Romano S, Erfani S, Bailey J. Normalized loss functions for deep learning with noisy labels
> > > > >
> > > > > [4] Zhou X, Liu X, Jiang J, Gao X, Ji X. Asymmetric loss functions for learning with noisy labels.
> > > > >
> > > > > [5] Wei J, Liu H, Liu T, Niu G, Sugiyama M, Liu Y. To smooth or not? when label smoothing meets noisy labels.
> > > > >
> > > > > [6] Xia X, Liu T, Han B, Gong C, Wang N, Ge Z, Chang Y. Robust early-learning: Hindering the memorization of noisy labels.

---

### Author Response · Authors · 2023-11-20
**Comparing with SOTA methods (ELR+ and DivideMix)**

Dear reviewers and AC,

Most reviewers have asked to compare SGN with the state-of-the-art methods of ELR+ and DivideMix. In the “Rationale for chosen baselines” paragraph in the response to Reviewer QVE6, we argued such a comparison is unfair as ELR+ and DivideMix have several add-ons only to improve performance, e.g., training two networks at the same time, averaging the predictions of the two networks at test time, and using a stronger data augmentation technique called mixup. To show the importance of these add-ons and the effectiveness of our approach, we propose to keep our original training setup exactly the same and **only use a single add-on in the form of mixup** (mixup alpha of 0.8). Despite only using a single add-on, SGN with mixup achieves higher top-1 accuracy on WebVision compared to both DivideMix and ELR+ (baseline results are gathered from [1]):

&nbsp;

| Method   	|   WebVision top-1	|  WebVision top-5  | ILSVRC12 top-1 |  ILSVRC12 top-5  |
| --------- |:-----:|:-----:|:-----:|:-----:|
| ELR   	|   76.26	|  91.26  	|   68.71	|   87.84	|
| SGN   	| 77.24  | 91.19  | 72.60  | 90.46  |
| ELR+ (uses mixup)  	| 77.78 | 91.68 | 70.29 | 89.76 |
| DivideMix (uses mixup) | 77.32 | 91.64 | 75.20 | 90.84 |
| SGN with mixup  	|  78.37 |  90.67 |  75.88 |   89.78	|

&nbsp;

For SGN with mixup, the validation accuracy continually improved, so we report the accuracies at the end of training rather than relying on early stopping. We believe the reviewers will be excited about these results, and we see no reason why more add-ons wouldn’t improve the performance further. We look forward to engaging in an active discussion about our individual responses (posted almost four days ago) and the presentation of these new impressive results.

Best regards,

Authors

**References**

[1] Liu S, Niles-Weed J, Razavian N, Fernandez-Granda C. Early-learning regularization prevents memorization of noisy labels.

---

> ### Comment · Reviewer_QVE6 · 2023-11-21
> **Question**
>
> Why are top-5 accuracy lower with mixup than without for both datasets? Do the authors have an intuition?

---

> ### Author Response · Authors · 2023-11-22
> **Discussion on the top-5 accuracy.**
>
> We appreciate that the reviewer initiated a discussion of our new results. We also noticed that the top-5 accuracy for SGN with mixup is lower than without it, but we had no clear intuition for why that was. Fortunately, we now have results with five different seeds that we can use to compare with the mean and standard deviation of SGN without mixup, which was also evaluated at the end of training.
>
> | Method/Comparison  	 |   WebVision top-1    |  WebVision top-5  | ILSVRC12 top-1 |  ILSVRC12 top-5  |
> | --------- |:-----:|:-----:|:-----:|:-----:|
> | SGN w/o mixup  	 | 76.12 $\pm$ 0.36  | 90.74 $\pm$ 0.29  | 72.72 $\pm$ 0.17  | 90.35 $\pm$ 0.28 |
> | SGN w/ mixup 	 |  78.21 $\pm$ 0.18 | 90.82 $\pm$ 0.31 |  75.53 $\pm$ 0.47  |  89.95 $\pm$ 0.21    |
> | mixup vs no mixup    |  +1 (<0.0001) | 0 (0.6846) | +1 (<0.0001)  | -1 (0.0339) |
>
> The last row is marked with 0 if the difference is not statistically significant (p $\geq$ 0.05), and +1 or -1 if it is, and the two-tailed P value is reported in parentheses.
>
> We find that there is an extremely significant difference in terms of top-1 accuracy in both cases. Interestingly, there is no significant difference in top-5 accuracy for the WebVision validation set, but there is, although barely, on the ImageNet validation set.
>
> In light of these findings, and noting the fact that top-5 accuracy is a much more forgiving metric, we attribute the differences in top-5 accuracy to inherent variance in the training and/or evaluation.
>
> Thanks again for your active participation in the discussion period!

---

### Meta-Review · Area_Chair_jv9o · 2023-12-09

**Metareview:**

The authors address the problem of overfitting in deep learning and propose a unification of loss reweighting and label correction to enhance robustness against label noise in classification tasks. To this end, they frame the original problem as a regression task, where loss reweighting and label correction can naturally be realised by means of a shifted Gaussian label noise model. The authors show that their method compares favourably to baselines.

Although the reviewers raise a number of critical points in their original reports, there is agreement that the paper has great potential and makes a significant contribution. The authors showed a high level of commitment during the rebuttal phase and did their best to respond to the comments and to improve the submission. This was appreciated and positively acknowledged by all. In the discussion between authors and reviewers, some critical points could be resolved and some questions clarified. Eventually, there is agreement that the paper has enough merit to be accepted. Just as an additional remark, the authors might be interested in label relaxation, a label correction technique that has recently been proposed as a generalisation of label smoothing.

**Justification For Why Not Higher Score:**

Good paper, but not outstanding.

**Justification For Why Not Lower Score:**

Good enough to be accepted.

---

### Decision · Program_Chairs · 2024-01-16

Accept (poster)